# PHyCLIP: $\ell_1$-Product of Hyperbolic Factors Unifies Hierarchy and Compositionality in Vision–Language Representation Learning

**Daiki Yoshikawa**[1] **& Takashi Matsubara**[1,2]
[1]Hokkaido University  [2]CyberAgent

## Abstract

Vision–language models have achieved remarkable success in multi-modal representation learning from large-scale pairs of visual scenes and linguistic descriptions. However, they still struggle to simultaneously express two distinct types of semantic structures: the hierarchy within a concept family (e.g., *dog $\preceq$ mammal $\preceq$ animal*) and the compositionality across different concept families (e.g., "a dog in a car" $\preceq$ *dog*, *car*). Recent works have addressed this challenge by employing hyperbolic space, which efficiently captures tree-like hierarchy, yet its suitability for representing compositionality remains unclear. To resolve this dilemma, we propose *PHyCLIP*, which employs an $\ell_1$-*P*roduct metric on a Cartesian product of *Hy*perbolic factors. With our design, intra-family hierarchies emerge within individual hyperbolic factors, and cross-family composition is captured by the $\ell_1$-product metric, analogous to a Boolean algebra. Experiments on zero-shot classification, retrieval, hierarchical classification, and compositional understanding tasks demonstrate that PHyCLIP outperforms existing single-space approaches and offers more interpretable structures in the embedding space.

## 1 Introduction

Vision–language models have become a central paradigm for learning transferable representations across visual and textual modalities. As exemplified by CLIP (Radford et al., 2021), contrastive pre-training maps images and texts to embeddings and enables strong zero-shot transfer on classification, retrieval, and related tasks. However, compressing the semantics of an instance into a single point makes it challenging to faithfully encode two semantic structures at once: *hierarchy* (*is-a* relations in a concept family) and *compositionality* (conjunction across distinct concept families).

Visual and linguistic concepts linked by *is-a* relations form tree-like taxonomic *hierarchies*. For example, a dog *is a* mammal, which in turn *is an* animal, as shown in the upper part of Fig. 1. Because the number of nodes grows exponentially with depth, Euclidean geometry struggles to faithfully represent such trees, whereas hyperbolic geometry aligns well with this growth (Bridson & Haefliger, 1999; Sarkar, 2011). These observations have motivated the development of hyperbolic embeddings (Nickel & Kiela, 2017) and hyperbolic entailment cones, which encode partial orders via inclusion (Ganea et al., 2018a). Within vision–language representation learning, MERU (Desai et al., 2023) and HyCoCLIP (Pal et al., 2025) leverage these approaches to capture image–text relations; for instance, an image of a dog *is an* instance of the linguistic concept dog (see the lower part of Fig. 1).

Beyond taxonomic structure, images and texts exhibit *compositionality*. For example, the description "a dog in a car" binds concepts dog and car from distinct concept families (animals and transportation), as shown in the middle part of Fig. 1. Classical approaches express composition via logical conjunction or additive operations (e.g., Boolean algebra, bag-of-words, and vector addition in word2vec) (Hinton et al., 1986; Mikolov et al., 2013; Vendrov et al., 2016), but these struggle to encode semantic hierarchy efficiently. Conversely, while hyperbolic geometry captures hierarchy, it lacks a canonical operation for composition. Möbius addition in hyperbolic spaces (Ungar, 2008) is not aligned with standard vector addition or Boolean structures (Higgins et al., 2018). Intersections of regions (such as hyperbolic entailment cones) can approximate conjunction but offer no general guarantees of representational efficiency for arbitrary co-occurrences.

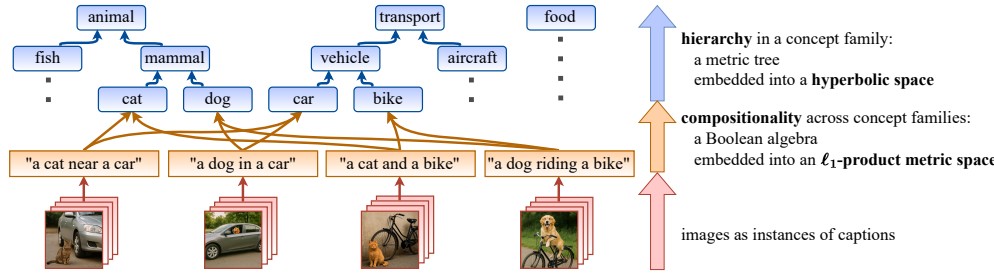

Figure 1: **Conceptual diagram of hierarchical and compositional structures.** While all arrows represent entailments ($\preceq$), they differ in nature. (upper) Linguistic concepts organize tree-like taxonomic *hierarchies* of concept families, each of which can be embedded into a hyperbolic space (Sarkar, 2011). (middle) Images and texts exhibit *compositionality* across distinct concept families, which can be captured by a Boolean algebra or an $\ell_1$-product metric. (lower) Images are instances of their corresponding captions.

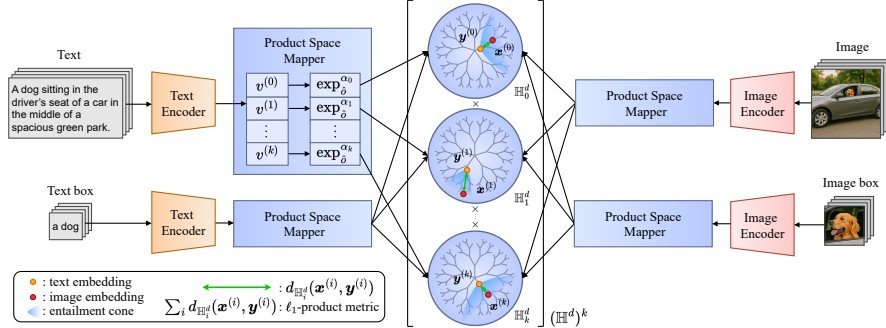

Figure 2: **Overview of PHyCLIP.** Images and texts are encoded as points $\boldsymbol{X}$ in an $\ell_1$-product metric space of hyperbolic factors, $(\mathbb{H}^d)^k$, that is, as tuples of points $\boldsymbol{x}^{(i)}$ in hyperbolic spaces $\mathbb{H}_i^d$, where their distance is defined by the sum of hyperbolic distances. The entailment relations $\boldsymbol{X} \preceq \boldsymbol{Y}$ are encoded using entailment cones as $\boldsymbol{x}^{(i)} \in C(\boldsymbol{y}^{(i)})$ within hyperbolic factors $\mathbb{H}_i^d$.

To resolve this dilemma, we propose *PHyCLIP*, which leverages an $\ell_1$-*P*roduct metric on a Cartesian product of *Hy*perbolic factors, as depicted in Fig. 2[1]. Our design follows two classical correspondences: (i) metric trees admit low-distortion embeddings into hyperbolic spaces, so hyperbolic factors embed *intra-family* taxonomies (Sarkar, 2011; Sala et al., 2018; van Spengler & Mettes, 2025); and (ii) finite Boolean algebras with the Hamming distance embed isometrically into an $\ell_1$ space, so an $\ell_1$-product metric naturally supports *cross-family* Boolean-like composition (Deza & Laurent, 1997). Intuitively, each bit for an atomic concept (e.g., dog, car, tomato) in the Boolean algebra is replaced with a hyperbolic factor for a concept family (e.g., animals, transportation, food), and the activation of multiple factors expresses composition (e.g., "dog and car"). Unlike previous mixed-curvature models (Gu et al., 2019; Wang et al., 2024; Gao et al., 2025), our space uses an $\ell_1$-product metric rather than a Riemannian ($\ell_2$) product metric and constrains each factor to have negative curvature. Our contributions are summarized as follows.

**Balancing Hierarchy and Compositionality.** We introduce PHyCLIP, a vision–language model that leverages an $\ell_1$-product metric space of hyperbolic factors to jointly capture *hierarchy* (within factors) and *compositionality* (across factors).

**Theoretical Support.** We formally link Boolean lattices to $\ell_1$-product metrics and metric trees to hyperbolic factors, explaining that an $\ell_1$-product metric space of hyperbolic factors aligns better with the dual semantic structures than standard metric spaces (e.g., Euclidean or hyperbolic).

**Superior Performance and Interpretability.** Experiments on zero-shot classification, image–text retrieval, hierarchical classification, and compositional understanding demonstrate that PHyCLIP achieves consistent improvements over baselines that use standard metric spaces. Visualizations find that intra-family taxonomies emerge within individual factors, and composing concepts leads to the simultaneous activation of the corresponding factors, analogous to a Boolean algebra.

---

[1]Codes are available at `https://github.com/tksmatsubara/PHyCLIP`.

## 2 THEORETICAL BACKGROUND AND MOTIVATION

**Geometry and Embedding of Hierarchies.** Concepts in natural language linked by *is-a* (hypernymy/hyponymy, generalization/specialization, entailment) relations form a partially ordered set (poset) and typically exhibit deep hierarchical structure. A poset $(P, \preceq)$ is a set equipped with an order relation $\preceq$ (which is reflexive, antisymmetric, and transitive). A typical example is dog $\preceq$ mammal $\preceq$ animal, where a dog *is a* type of mammal; equivalently, if an entity is a dog, then this *entails* that the entity is a mammal. Large lexical resources such as WordNet provide such relations in the form of a directed acyclic graph with multiple inheritance (e.g., dog $\preceq$ domestic animal) (Miller, 1995). For modeling or computational convenience, many studies approximate this hierarchy with a taxonomic tree (Morin & Bengio, 2005; Mnih & Hinton, 2008). The distance between two nodes (i.e., words) in a tree is often defined as the length of their shortest path, inducing a type of *metric tree*. The distance of a node from the root is a natural measure of specialization of the concept that the node represents. The root node specifies the most general concept (e.g., entity) and effectively means nothing (i.e., the absence). See technical details in Appendix A.

**Theorem 1** (Hyperbolic embedding of trees (Sarkar, 2011))**.** *Let $\mathbb{H}^d$ be a $d$-dimensional hyperbolic space with the hyperbolic distance $d_{\mathbb{H}^d}$. For every finite metric tree $T$ (and every infinite metric tree $T$ with known bounds for maximum degree and minimum edge length), and for every $\varepsilon > 0$, there exist a scale $\tau > 0$ and an embedding $f : \tau T \to \mathbb{H}^2$ such that the distortion is at most $1 + \varepsilon$; that is, there exists a $(1 + \varepsilon, 0)$-quasi-isometric embedding $f$ up to scaling.*

See Theorem 5 in Sarkar (2011) for the proof. This explains the empirical success of hyperbolic embeddings for hierarchical data (Nickel & Kiela, 2017; 2018; Ganea et al., 2018a; Sala et al., 2018; Tifrea et al., 2019; van Spengler & Mettes, 2025). In practice, $\mathbb{H}^d$ with $d > 2$ is common for achieving better performance.

**Geometry and Embedding of Compositionality.** Beyond taxonomic structure, images and texts often exhibit compositionality: they mention multiple concepts to indicate the co-occurrence or conjunction of those concepts. For example, the description "a dog in a car" mentions concepts dog and car. Such data suggest another type of entailment relation, as an image of "a dog in a car" can be regarded as an image of dog as well as an image of car. The resulting structure is no longer a tree but rather a more general poset. While hyperbolic embeddings remain an option, it is natural to explore alternatives that more directly capture compositionality.

Order embeddings $(\mathbb{R}^n, \preceq)$ (Vendrov et al., 2016) assign each concept a point $\boldsymbol{x} \in \mathbb{R}^n$ and declare $\boldsymbol{x} \preceq \boldsymbol{y}$ iff $x_i \geq y_i$ for all coordinates $i$. This is equivalent to the inclusion relation between associated upper orthants $U(\boldsymbol{x}) := \{\boldsymbol{z} \in \mathbb{R}^n : z_i \geq x_i \, \forall i\}$, i.e., $\boldsymbol{x} \preceq \boldsymbol{y}$ iff $U(\boldsymbol{x}) \subseteq U(\boldsymbol{y})$. Then, the coordinate-wise $\max$ (i.e., the union of orthants) expresses conjunction (e.g., $\max(\mathsf{dog}, \mathsf{car})$ includes "a dog in a car"), and the coordinate-wise $\min$ yields shared concepts (e.g., $\min$("a dog in a car", "a dog on a sofa") $\approx$ dog). Similarly, box embeddings use axis-aligned hyperrectangles in $\mathbb{R}^n$ (Vilnis et al., 2018; Li et al., 2019; Dasgupta et al., 2020). In hyperbolic space, hyperbolic entailment cones use geodesic conical regions (Ganea et al., 2018a), and disk embeddings use hyperballs (Suzuki et al., 2019). Compared with hyperbolic embeddings for pure hierarchies, there has been less theoretical analysis of these region-based embeddings for compositionality. Our work extends this line to capture hierarchy and compositionality simultaneously.

**Boolean Lattice and Its Relation to Order Embedding.** In an *is-a* taxonomy, any two nodes have at least one common generalization, whereas they need not share a common specialization. A *lattice* is a poset in which any two nodes have both a common generalization (join) and a common specialization (meet). Consider $n$ atomic concepts $\mathcal{C} = \{c_1, \ldots, c_n\}$ (e.g., $\{\mathsf{dog}, \mathsf{car}, \mathsf{tomato}, \ldots \}$). A subset $S \subseteq \mathcal{C}$ denotes the conjunction of the concepts specified in $S$, and the inclusion relation $S \supseteq T$ (e.g., $\{\mathsf{dog}, \mathsf{car}\} \supseteq \{\mathsf{dog}\}, \{\mathsf{car}\}$) induces the order relation $S \preceq T$ (e.g., $\{\mathsf{dog}, \mathsf{car}\} \preceq \{\mathsf{dog}\}, \{\mathsf{car}\}$). In this way, the *Boolean lattice* $(2^{\mathcal{C}}, \subseteq)$ over all such subsets naturally represents the compositionality of atomic concepts as a non-taxonomic poset. When focusing on operations rather than order, it is also referred to as a Boolean algebra. At the same time, using an indicator $\chi : 2^{\mathcal{C}} \to \{0, 1\}^n$, the Boolean lattice can be regarded as a metric space $(\{0, 1\}^n, d_{\mathrm{Ham}})$ with the Hamming distance. See Appendix A and Ganter & Wille (1999); Davey & Priestley (2002) for more details.

**Definition 1** ($\ell_p$-product metric space). *Let $\{(X_i, d_{X_i})\}_{i=1}^k$ be non-trivial metric spaces. An $\ell_p$-product metric space of $\{(X_i, d_{X_i})\}_{i=1}^k$ is a Cartesian product space $\prod_{i=1}^k X_i$ equipped with the $\ell_p$-product metric $d_p((\boldsymbol{x}^{(1)}, \dots, \boldsymbol{x}^{(k)}), (\boldsymbol{y}^{(1)}, \dots, \boldsymbol{y}^{(k)})) = (\sum_{i=1}^k |d_{X_i}(\boldsymbol{x}^{(i)}, \boldsymbol{y}^{(i)})|^p)^{1/p}.$*

**Proposition 1** (Embedding of Boolean Lattice). *A Boolean lattice $(2^{\mathcal{C}}, \preceq)$ for $n$ atomic concepts can be embedded into the poset $(\mathbb{R}^n, \preceq)$ used by order embeddings while preserving the order relations. As a metric space $(\{0,1\}^n, d_{\mathrm{Ham}})$, it is isometrically embedded into an $\ell_1$-product metric space $(\prod_{i=1}^k X_i, d_1)$ for any $k \geq n$ after appropriate per-factor scaling. However, it admits no isometric embedding into a hyperbolic space $\mathbb{H}^d$ for any $d \geq 2$ and $n \geq 2$.*

See Appendix B for the proof. The first part of the proposition explains that order embeddings can be regarded as a continuous relaxation of Boolean lattices. A pure Boolean lattice has remarkable expressivity for compositionality, but it is often too coarse as it considers combinations of all atomic concepts. Order embeddings enrich it by replacing each bit $\{0,1\}$ with $\mathbb{R}$. The remaining part explains the affinity between Boolean lattices and the $\ell_1$-product metric, as well as the incompatibility between Boolean lattices and hyperbolic spaces.

Note that we can also consider a weighted version of an $\ell_p$-product metric, where the distances on different factors can be weighted differently. However, in the context of isometric embeddings, these weights and per-factor scalings are essentially equivalent, as one can be absorbed into the other.

## 3 PHYCLIP AND ITS LOSS FUNCTIONS

**Embedding into an $\ell_1$-Product Metric Space of Hyperbolic Factors.** Here, we model the semantics encoded in images and texts as a conjunction of multiple atomic concepts each of which is taken from a distinct concept family (e.g., animals, transportation, food). This model is realized by replacing each bit $\{0,1\}$ of a Boolean lattice with a metric tree $T_i$ for a concept family. For example, the description "a dog in a car" is represented by a pair of nodes in metric trees $T_1$ and $T_2$ that encode *is-a* taxonomies of animals (e.g., dog $\preceq$ mammal $\preceq$ animal) and transportation (e.g., car $\preceq$ vehicle $\preceq$ transport), respectively. Notably, a *single* hyperbolic space cannot capture this product geometry (see Proposition 2 in Appendix B), whereas an $\ell_1$-product metric space of hyperbolic factors can.

**Theorem 2** (Embedding into an $\ell_1$-product metric space of hyperbolic factors). *Let $T_1, \dots, T_k$ be finite metric trees (or infinite metric trees with known bounds for maximum degree and minimum edge length) with metrics $d_{T_1}, \dots, d_{T_k}$. For every $\varepsilon > 0$, there exists a $(1+\varepsilon, 0)$-quasi-isometric embedding from the $\ell_1$-product metric space of these metric trees, $(\prod_{i=1}^k T_i, d_1)$, into an $\ell_1$-product metric space of $k$ two-dimensional hyperbolic factors, $((\mathbb{H}^2)^k, d_1)$, after appropriate per-factor scaling.*

Given the above, we propose embeddings into an $\ell_1$-product metric space of $k$ copies of $d$-dimensional hyperbolic factors $\mathbb{H}^d$, $((\mathbb{H}^d)^k, d_1)$. The total dimension of $(\mathbb{H}^d)^k$ is $kd$. Each hyperbolic factor $\mathbb{H}_i^d$ is intended to represent the taxonomy of a concept family, as well as aspects of inter-object relations (e.g., "a dog riding on something", "a car loading something"). An instance is embedded as a tuple $\boldsymbol{X} = (\boldsymbol{x}^{(1)}, \dots, \boldsymbol{x}^{(k)})$ for $\boldsymbol{x}^{(i)} \in \mathbb{H}_i^d$. Within each factor $\mathbb{H}_i^d$, we use standard hyperbolic embeddings (Nickel & Kiela, 2017) together with hyperbolic entailment cones (Ganea et al., 2018a) to encode *intra-family* hierarchy and image–text entailment, while *cross-family* compositionality is captured by the additive geometry of the $\ell_1$-product metric space.

**PHyCLIP for Vision–Language Representation Learning.** Here, we propose PHyCLIP for vision–language representation learning, depicted in Fig. 2. Let $I$ and $T$ denote instances of images and texts, respectively. From an instance, a $kd$-dimensional feature vector is produced, which is then sliced into $k$ segments $\boldsymbol{v}^{(i)}$ of dimension $d$ for $i = 1, \dots, k$, and each segment $\boldsymbol{v}^{(i)}$ is lifted via the exponential map to its corresponding hyperbolic factor $\mathbb{H}_i^d$ as $\boldsymbol{x}^{(i)}$, yielding the embedding $\boldsymbol{X} = (\boldsymbol{x}^{(1)}, \dots, \boldsymbol{x}^{(k)}) \in (\mathbb{H}^d)^k$. We denote the embeddings of the image $I$ and text $T$ by $\boldsymbol{I}$ and $\boldsymbol{T}$, respectively. Let $B$ denote the index set of instances in a mini-batch; we write the mini-batch of images as $\{\boldsymbol{I}_b\} = \{\boldsymbol{I}_b\}_{b \in B}$ for brevity.

An image $I$ is typically more specific than its corresponding text $T$ ($I \preceq T$) as the text $T$ may ignore some details of the image $I$. Following HyCoCLIP (Pal et al., 2025), we suppose that the

training data are enriched with box information: the image boxes $I^{\text{box}}$ are object-level crops of the original images $I$, and the text boxes $T^{\text{box}}$ are the corresponding nouns/phrases within the text $T$ ($I^{\text{box}} \preceq T^{\text{box}}$). An image box $I^{\text{box}}$ and a text box $T^{\text{box}}$ are more general than the full image $I$ and the full text $T$ ($I \preceq I^{\text{box}}$, $T \preceq T^{\text{box}}$), respectively, since they omit objects and words outside the boxes.

We will introduce the contrastive loss $\mathcal{L}_{\text{cont}}$ and entailment loss $\mathcal{L}_{\text{ent}}$, and the final objective is their sum weighted by a hyperparameter $\gamma$:

$$\mathcal{L}_{\text{overall}} = \mathcal{L}_{\text{cont}} + \gamma \mathcal{L}_{\text{ent}}. \tag{1}$$

**Contrastive Loss.** To represent each hyperbolic factor $\mathbb{H}_i^d$, we adopt the Lorentz model with a learnable curvature $-\alpha_i$ (Cannon et al., 1997; Nickel & Kiela, 2018; Lee, 2018). See Appendix C for implementation details. Following Definition 1, we define the distance on the $\ell_1$-product metric space $(\mathbb{H}^d)^k$ and its averaged version as

$$d_1(\boldsymbol{X}, \boldsymbol{Y}) = \sum_{i=1}^{k} d_{\mathbb{H}_i^d}(\boldsymbol{x}^{(i)}, \boldsymbol{y}^{(i)}), \quad d_{\text{avg}}(\boldsymbol{X}, \boldsymbol{Y}) = \tfrac{1}{k} d_1(\boldsymbol{X}, \boldsymbol{Y}). \tag{2}$$

To pull an embedding $\boldsymbol{X}_b$ close to its positive pair $\boldsymbol{Y}_b$ while pushing it away from negatives $\boldsymbol{Y}_a$ for $a \neq b$, we use the standard InfoNCE loss (Radford et al., 2021; Desai et al., 2023; Pal et al., 2025):

$$\mathcal{L}_{\text{cont}}(\{\boldsymbol{X}_b\}, \{\boldsymbol{Y}_b\}) = - \sum_{b \in B} \log \frac{\exp(-d_{\text{avg}}(\boldsymbol{X}_b, \boldsymbol{Y}_b)/\tau)}{\sum_{a \in B} \exp(-d_{\text{avg}}(\boldsymbol{X}_b, \boldsymbol{Y}_a)/\tau)} \tag{3}$$

where $\tau$ is a learnable temperature parameter. We average this loss over known pairs:

$$\mathcal{L}_{\text{cont}} = \tfrac{1}{4}(\mathcal{L}_{\text{cont}}(\{\boldsymbol{I}_b\}, \{\boldsymbol{T}_b\}) + \mathcal{L}_{\text{cont}}(\{\boldsymbol{T}_b\}, \{\boldsymbol{I}_b\}) + \mathcal{L}_{\text{cont}}(\{\boldsymbol{I}_b^{\text{box}}\}, \{\boldsymbol{T}_b^{\text{box}}\}) + \mathcal{L}_{\text{cont}}(\{\boldsymbol{T}_b^{\text{box}}\}, \{\boldsymbol{I}_b^{\text{box}}\})). \tag{4}$$

**Entailment Loss.** We also employ hyperbolic entailment cones to capture the entailment relations (Ganea et al., 2018a). See Appendix C for implementation details. For every point $\boldsymbol{y}^{(i)}$ in each hyperbolic factor $\mathbb{H}_i^d$, we define a geodesic conical region $C(\boldsymbol{y}^{(i)})$ with apex at $\boldsymbol{y}^{(i)}$ and half-aperture $\omega(\boldsymbol{y}^{(i)})$, where all points $\boldsymbol{x}^{(i)} \in C(\boldsymbol{y}^{(i)})$ are considered more specific than $\boldsymbol{y}^{(i)}$ (i.e., $\boldsymbol{x}^{(i)} \preceq \boldsymbol{y}^{(i)}$). Then, $\boldsymbol{x}^{(i)} \in C(\boldsymbol{y}^{(i)})$ iff $\phi(\boldsymbol{x}^{(i)}, \boldsymbol{y}^{(i)}) < \omega(\boldsymbol{y}^{(i)})$ for the exterior angle $\phi(\boldsymbol{x}^{(i)}, \boldsymbol{y}^{(i)})$. To penalize the violation of the inclusion relation $\boldsymbol{x}^{(i)} \in C(\boldsymbol{y}^{(i)})$ for a pair $(\boldsymbol{x}^{(i)}, \boldsymbol{y}^{(i)})$ such that $\boldsymbol{x}^{(i)} \preceq \boldsymbol{y}^{(i)}$, the entailment loss $L_{\text{ent}}$ is calculated as

$$L_{\text{ent},i}(\boldsymbol{X}, \boldsymbol{Y}) = \max(0, \phi(\boldsymbol{x}^{(i)}, \boldsymbol{y}^{(i)}) - \eta\omega(\boldsymbol{y}^{(i)})), \quad L_{\text{ent}}(\boldsymbol{X}, \boldsymbol{Y}) = \tfrac{1}{k} \sum_{i=1}^{k} L_{\text{ent},i}(\boldsymbol{X}, \boldsymbol{Y}), \tag{5}$$

where hyperparameter $\eta$ controls the margin (Pal et al., 2025). We sum this loss over known pairs:

$$\mathcal{L}_{\text{ent}} = \sum_{b \in B} \left( L_{\text{ent}}(\boldsymbol{I}_b, \boldsymbol{T}_b) + L_{\text{ent}}(\boldsymbol{I}_b^{\text{box}}, \boldsymbol{T}_b^{\text{box}}) + L_{\text{ent}}(\boldsymbol{I}_b, \boldsymbol{I}_b^{\text{box}}) + L_{\text{ent}}(\boldsymbol{T}_b, \boldsymbol{T}_b^{\text{box}}) \right). \tag{6}$$

**Discussions.** PHyCLIP requires $k$ evaluations of $\cosh$ and $\sinh$ for the exponential maps and $k$ evaluations of $\operatorname{arcosh}$ for factor-wise distances. These operations are fully parallelized across factors and are needed only once per embedding or distance. In practice, the wall-clock time is dominated by the Vision Transformer and text encoder, and the additional cost from these operations is negligible at the scale of our backbones. PHyCLIP introduces only $k = 64$ additional parameters to learn curvatures, which is negligible compared to the 86M parameters for the base Vision Transformer.

When the negative curvature $\alpha_i$ of a hyperbolic factor $\mathbb{H}_i^d$ is multiplied by $1/c^2$, the Riemannian metric is scaled by $c^2$, and the distance between any two points becomes $c$ times larger. Hence, learning the negative curvature effectively corresponds to learning the combination weight of a weighted $\ell_1$-product metric. Since the distance of each instance's embedding from the origin can be adjusted independently for each instance, a small curvature does not necessarily mean that the corresponding factor is emphasized. If all embedding points lie near the origin, that factor is effectively down-weighted.

## 4 EXPERIMENTS

### 4.1 TRAINING DETAILS

**Datasets.** We trained all models on the Grounded Image–Text Pairs (GRIT) dataset (Peng et al., 2023), which consists of automatically annotated image–text pairs with bounding boxes and corresponding nouns/phrases. Although the dataset is documented to contain 20.5 million pairs with 35.9 million box annotations, we were able to obtain 14.0 million pairs with 26.6 million box annotations due to outdated public links. This scale remains considerably larger than manually annotated resources such as Flickr30K Entities (Plummer et al., 2015).

Table 1: Zero-shot image classification.

| | w/ boxes | General datasets | | | | | | Fine-grained datasets | | | | | | Specialized datasets | | | |
|---|---|---|---|---|---|---|---|---|---|---|---|---|---|---|---|---|---|
| | | ImageNet | CIFAR-10 | CIFAR-100 | SUN397 | Caltech-101 | STL-10 | Food-101 | CUB | Cars | Aircraft | Pets | Flowers | DTD | EuroSAT | RESISC45 | Country211 |
| CLIP | | 38.87 | 76.26 | 48.19 | 50.70 | 73.62 | 93.03 | 51.19 | 12.90 | 7.82 | 3.01 | 45.89 | 21.16 | 22.02 | 35.73 | 42.03 | 5.13 |
| CLIP | ✓ | 38.81 | 76.53 | 48.59 | 50.80 | 74.29 | 93.34 | 51.05 | 12.70 | 8.40 | 2.89 | 46.19 | 21.32 | 21.74 | 37.49 | 41.78 | 5.10 |
| MERU | | 37.96 | 77.63 | 46.37 | 49.39 | 72.10 | 93.14 | 51.67 | 11.09 | 7.80 | 3.53 | 43.36 | 19.98 | 22.18 | 38.81 | 41.77 | 4.86 |
| MERU | ✓ | 38.08 | 78.14 | 46.80 | 49.59 | 72.69 | 93.28 | 51.92 | 10.70 | 7.77 | 3.53 | 43.22 | 18.31 | 22.07 | 37.31 | 41.73 | 5.01 |
| HyCoCLIP | ✓ | 43.80 | 89.00 | 58.59 | 54.49 | 76.14 | 94.96 | 52.64 | 14.90 | 10.24 | 3.57 | 53.33 | 19.41 | 25.90 | 36.36 | 46.97 | 5.64 |
| **PHyCLIP** | ✓ | **44.31** | 89.33 | 59.05 | 55.32 | 76.35 | 94.84 | 57.26 | 15.90 | 10.89 | 3.24 | 54.18 | 19.98 | 25.50 | 36.29 | 48.22 | 5.56 |

The best and second-best performances are emphasized by bold fonts and underlines, respectively.

Table 2: Zero-shot retrieval and hierarchical classification.

| | w/ boxes | Text → Image | | | | Image → Text | | | | Hierarchical Classification | | | | |
|---|---|---|---|---|---|---|---|---|---|---|---|---|---|---|
| | | COCO | | Flickr | | COCO | | Flickr | | WordNet | | | | |
| | | R@5 | R@10 | R@5 | R@10 | R@5 | R@10 | R@5 | R@10 | TIE($\downarrow$) | LCA($\downarrow$) | $J(\uparrow)$ | $P_H(\uparrow)$ | $R_H(\uparrow)$ |
| CLIP | | 56.29 | 67.53 | 83.15 | 89.58 | 70.32 | 80.09 | **91.60** | 95.60 | 3.750 | 2.276 | 0.7774 | 0.8471 | 0.8483 |
| CLIP | ✓ | 56.20 | 67.50 | 82.75 | 89.42 | 70.35 | 80.19 | 91.10 | 95.63 | 3.736 | 2.279 | 0.7784 | 0.8473 | 0.8501 |
| MERU | | 55.73 | 67.02 | 82.15 | 89.05 | 69.57 | 79.33 | 90.77 | **95.83** | 3.815 | 2.294 | 0.7733 | 0.8454 | 0.8450 |
| MERU | ✓ | 55.87 | 67.21 | 81.96 | 88.89 | 69.70 | 79.69 | 91.20 | **95.83** | 3.802 | 2.289 | 0.7740 | 0.8457 | 0.8455 |
| HyCoCLIP | ✓ | 57.11 | 68.32 | 83.06 | 89.63 | 69.51 | 79.73 | 91.47 | 95.63 | 3.319 | 2.092 | 0.8043 | 0.8676 | 0.8661 |
| **PHyCLIP** | ✓ | **58.03** | **69.05** | **83.39** | **89.93** | **70.94** | **80.86** | 91.20 | 95.53 | **3.294** | **2.083** | **0.8059** | **0.8684** | **0.8672** |

**Baselines.** We compare PHyCLIP with CLIP (Radford et al., 2021), MERU (Desai et al., 2023), and HyCoCLIP (Pal et al., 2025). CLIP is a seminal vision–language model trained with contrastive learning based on the cosine similarity. MERU extends CLIP by lifting embeddings to hyperbolic space and using hyperbolic entailment cones to represent hierarchy. HyCoCLIP further leverages box annotations to better capture intra-modal hierarchy. All models were trained from scratch on GRIT for fair comparison, and we report the average scores over three runs with fixed random seeds. For PHyCLIP, we set the number of factors to $k = 64$ and the dimension of each factor to $d = 8$, resulting in a total dimension of $512$. We followed the training protocols and hyperparameters used in the official implementations of HyCoCLIP (Pal et al., 2025); see Appendix C for details. We report results obtained with the base Vision Transformer as an image encoder (Dosovitskiy et al., 2021). Supplementary results are provided in Appendix D.1.

## 4.2 EXPERIMENTAL RESULTS

**Zero-shot Image Classification.** We evaluated the geometry of embedding space via zero-shot image classification, following the protocol standardized by CLIP (Radford et al., 2021). Images are classified using the similarity to the averaged embedding of template text queries for classes across 16 datasets, grouped into General, Fine-grained, and Specialized. General datasets cover broad, heterogeneous concept families (e.g., animals, transportation, household objects). Fine-grained datasets focus on visually similar subclasses within a single concept family (e.g., specific food, dog breeds). Specialized datasets are domain-specific (e.g., texture images, satellite imagery).

Table 1 summarizes top-1 accuracies. PHyCLIP obtained consistent performance gains, particularly on General datasets, which we attribute to assigning concept families to hyperbolic factors that naturally support coarse-grained classifications. Within Fine-grained datasets, PHyCLIP achieved remarkable improvements on Food-101 (Bossard et al., 2014) and Oxford-IIIT Pets (Parkhi et al., 2012), implying that it also learned intra-family taxonomies without confusion with other families. Although not best on every dataset, the performance gap on FGVC-Aircraft (Maji et al., 2013) is small, and the specialized datasets are out-of-distribution relative to GRIT. Consistent with prior findings (Pal et al., 2025), CLIP and MERU do not yield clear improvements with box annotations. Overall, PHyCLIP is the strongest zero-shot classifier among the comparison models.

**Zero-shot Image and Text Retrieval.** We evaluate cross-modal alignment via zero-shot retrieval in the shared embedding space: given a text query, retrieve the nearest images, and vice versa. This is also a standard benchmark for vision–language models (Radford et al., 2021). We used the COCO validation set (Lin et al., 2014) and the Flickr30K test set (Young et al., 2014; Karpathy & Li, 2015).

Table 3: Compositional understanding through hard-negative classification.

| | w/ boxes | VL-CheckList–Object | | | | | | SugarCrepe | | | | | | | |
| | | Location | | | Size | | | Replace | | | Swap | | Add | | |
| | | Center | Mid | Margin | Large | Medium | Small | Obj | Att | Rel | Obj | Att | Obj | Att | Overall |
| CLIP | | 61.90 | 60.30 | 60.40 | 63.87 | 60.17 | 58.23 | 89.37 | 79.95 | 69.54 | 60.54 | 66.02 | 80.39 | 73.36 | 77.72 |
| CLIP | ✓ | 61.93 | 59.33 | 60.83 | 63.70 | 60.80 | 58.07 | 89.69 | 80.33 | 69.49 | **61.63** | **66.47** | 80.62 | 73.55 | 77.97 |
| MERU | | 61.27 | 59.03 | 58.97 | 63.97 | 57.70 | 56.07 | 89.10 | 80.50 | 69.44 | 60.82 | 65.32 | 80.47 | 74.90 | 77.81 |
| MERU | ✓ | 61.03 | 58.47 | 58.73 | 62.63 | 58.70 | 56.47 | 89.39 | 79.95 | 69.65 | 60.41 | 66.07 | 80.41 | **75.34** | 77.93 |
| HyCoCLIP | ✓ | 70.43 | 69.50 | 67.80 | 72.57 | 66.13 | 67.20 | **91.38** | 79.74 | 67.24 | 54.69 | 63.66 | 82.57 | 74.23 | 77.99 |
| **PHyCLIP** | ✓ | **71.20** | **70.30** | **70.37** | **73.73** | **68.10** | **67.83** | 91.06 | **81.05** | 66.36 | 57.41 | 65.87 | **83.24** | 73.80 | **78.32** |

We report Recall at $k$ (R@$k$), the fraction of queries for which the paired instance appears in the top-$k$ retrieved results.

Results are summarized in the left half of Table 2. PHyCLIP achieves the best performance across almost combinations of performance measures and datasets, which supports our choice of the $\ell_1$-product metric in Eq. (2). This metric sums distances over hyperbolic factors; when an object specified in the text is absent from a candidate image, or an unspecified object is present, the corresponding factor incurs a large penalty. By contrast, a single hyperbolic space implicitly encodes the presence or absence of objects as hierarchical relations, which may weaken penalties for such mismatches and hinder separability of hard negatives. All models perform competitively on the text retrieval for Flickr, suggesting the performance saturation.

**Hierarchical Classification.** We evaluate the expressivity for the *is-a* taxonomy via hierarchical classification (Kosmopoulos et al., 2015; Pal et al., 2025) on ImageNet (Russakovsky et al., 2015), where class labels are enriched by WordNet (Miller, 1995) and errors between predicted and ground-truth classes are measured on the WordNet graph with unit-length edges: Tree Induced Error (TIE) is their graph distance; Lowest Common Ancestor (LCA) error is the maximum of the distances to their LCA; Jaccard similarity $J$, hierarchical precision $P_H$, and hierarchical recall $R_H$ are similarities between the sets of ancestors.

Results are summarized in the right half of Table 2. PHyCLIP achieves superior scores across all metrics, indicating not only higher classification accuracy but also that misclassifications tend to be close to the ground-truth class in the taxonomy. By handling cross-family compositionality via the $\ell_1$-product metric, each hyperbolic factor can devote capacity to a cleaner intra-family *is-a* taxonomy, thereby yielding disentangled, hierarchy-aligned representations.

**Compositional Understanding.** We assess the expressivity of compositionality via hard negative classification using VL-CheckList (Zhao et al., 2022) and SugarCrepe (Hsieh et al., 2023). Both benchmarks provide a kind of image-to-text retrieval task that require models to distinguish ground-truth captions from hard negatives created by altering objects, attributes, or relations in the ground-truth captions. Following Pal et al. (2025), we evaluate the Object subset of VL-CheckList, in which a noun for a single object in each caption is randomly replaced. The results are summarized by the replaced object's location (center/mid/margin) and size (small/medium/large) in the image. We also evaluate all seven subsets in SugarCrepe, in which objects, attributes, and relations are replaced, swapped, or added in each caption.

As shown in Table 3, PHyCLIP yields a substantial improvement on VL-CheckList–Object. It successfully represents object presence robustly with respect to location and size. On SugarCrepe, PHyCLIP obtains the best average score, which exceeds that of the second-best model by a large margin, whereas the other models cluster within a small range. Performance on attribute replacement and swapping is robust, suggesting that our design decouples intra-family taxonomy from cross-family composition and thereby emphasizes attribute–object binding. By contrast, we observe modest drops in relation replacement and object swapping, which implies that our design is less sensitive to inter-object relations, potentially because of its Boolean-like nature. This suggests that even within image-to-text retrieval, models exhibit distinct strengths and weaknesses depending on the type of compositions. Consequently, the performance variation observed in image-to-text retrieval on COCO and Flickr in Table 2 is likely attributed to which types of hard negatives, and in what proportion, are included in each benchmark.

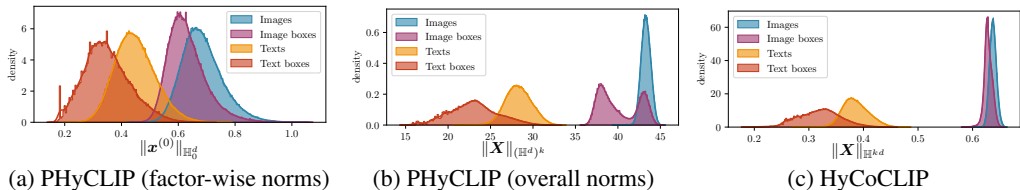

(a) PHyCLIP (factor-wise norms)  (b) PHyCLIP (overall norms)  (c) HyCoCLIP

Figure 3: **Norm distributions.** In (b) and (c), image norms are consistently larger than text norms, because images are more specific than their paired texts ($I_b \preceq T_b$). However, in a single hyperbolic factor shown in (a), image and text norms largely overlap, as PHyCLIP may keep some factors unused for instances that do not contain the corresponding concept families.

**Ablation Study.** We investigate the contributions of embedding space factorization and the $\ell_1$-product metric through ablation studies, summarized in Table 4. We fix the total embedding dimension $kd$ and vary the number of factors, $k$. When $k = 1$ (equivalent to HyCoCLIP), performance is the lowest on most measures; increasing $k$ generally improves results, except for text retrieval, thereby demonstrating the benefit of factorization. Most performance measures peak at $k = 64$ or $k = 128$, although zero-shot classification accuracy for Food-101 (Bossard et al., 2014) drops substantially at $k = 128$, indicating

Table 4: Ablation study.

| # of factors, $k$ | # of dims., $d$ | product metric | curvature | classification ImageNet | classification Food-101 | retrieval COCO, R@5 Image | retrieval COCO, R@5 Text | hierarchical TIE | hierarchical J |
|---|---|---|---|---|---|---|---|---|---|
| 1 | 512 | – | hyp. | 43.80 | 52.64 | 57.11 | 69.51 | 3.319 | 0.8043 |
| 8 | 64 | $\ell_1$ | hyp. | 44.38 | 54.61 | 57.80 | 70.80 | 3.273 | 0.8072 |
| 16 | 32 | $\ell_1$ | hyp. | 44.09 | 55.29 | 57.26 | 69.22 | 3.287 | **0.8066** |
| 32 | 16 | $\ell_1$ | hyp. | 43.90 | 54.48 | 56.70 | 66.92 | 3.324 | 0.8035 |
| 64 | 8 | $\ell_1$ | hyp. | **44.31** | **57.26** | **58.03** | 70.94 | 3.294 | 0.8059 |
| 128 | 4 | $\ell_1$ | hyp. | 44.16 | 53.96 | 57.79 | **71.18** | **3.284** | 0.8064 |
| 64 | 8 | $\ell_2$ | hyp. | 43.32 | 53.39 | 57.09 | 70.53 | 3.367 | 0.8011 |
| 64 | 8 | $\ell_\infty$ | hyp. | 6.55 | 10.33 | 8.77 | 14.51 | 9.697 | 0.4247 |
| - | - | $\ell_2$ | mixed | 39.34 | 49.05 | 56.72 | 70.81 | 3.712 | 0.7797 |

that overly fine factorization may impair the representation of intra-family taxonomy. Replacing the $\ell_1$-product metric with the Riemannian ($\ell_2$) or $\ell_\infty$-product metric consistently degrades performance. We also investigated the mixed-curvature model of the Euclidean, hyperbolic, and spherical spaces of 256, 128, and 128 dimensions, respectively (Gu et al., 2019; Wang et al., 2024; Gao et al., 2025), and found an overall inferior performance. These results support that the $\ell_1$-product metric provides a more effective way to aggregate cross-family composition.

### 4.3 VISUALIZATIONS OF HYPERBOLIC FACTORS

**Norm Distributions.** Figure 3 plots the empirical distributions of embedding norms. (b) and (c) show that, in both PHyCLIP and HyCoCLIP, image norms are consistently larger than text norms and are tightly concentrated. These models consider images to be more specific than their paired texts, $I_b \preceq T_b$, which encourages the image embedding $I_b$ to lie within the text's hyperbolic entailment cone $C(T_b)$ (i.e., $I_b \in C(T_b)$), yielding larger image norms. However, (a) shows that, within individual hyperbolic factors of PHyCLIP, the image and text distributions largely overlap and are broadly dispersed. This is because instances without a particular concept family lie near the origin in the corresponding factor, in other words, do not use unrelated factors. Consequently, PHyCLIP leverages a broader portion of the embedding space and facilitates meaningful distances and taxonomic structures.

**Intra-Family Hierarchies in Hyperbolic Factors.** We obtain factor-wise embeddings $x^{(i)}$ of single-concept prompts (e.g., "a photo of a dog" and "a photo of a car") and summarize their norms $\|x^{(i)}\|_{\mathbb{H}_i^d}$ across $k = 64$ factors in Fig. 4 (a). The "dog" embedding exhibits its largest norm in factor $i = 39$ while remaining near the origin in factor $i = 9$. Conversely, the "car" embedding peaks in factor $i = 9$ and is suppressed in factor $i = 39$. We also visualize the top $0.1\%$ of GRIT images by the embedding norm for each factor. Factor $i = 39$ yields various mammals, whereas factor $i = 9$ shows vehicles and everyday-carry items.

Figure 5 visualizes embeddings of various single-concept prompts projected by HoroPCA (Chami et al., 2021) from $d = 8$-dimensional hyperbolic factors onto 2D disks. Embeddings of mammal-related terms are spread over a wide area in factor $i = 39$. Dog-related terms cluster in the left half, cat-related terms in the right, and "chihuahua," "corgi," and "puppy" are positioned farther from the

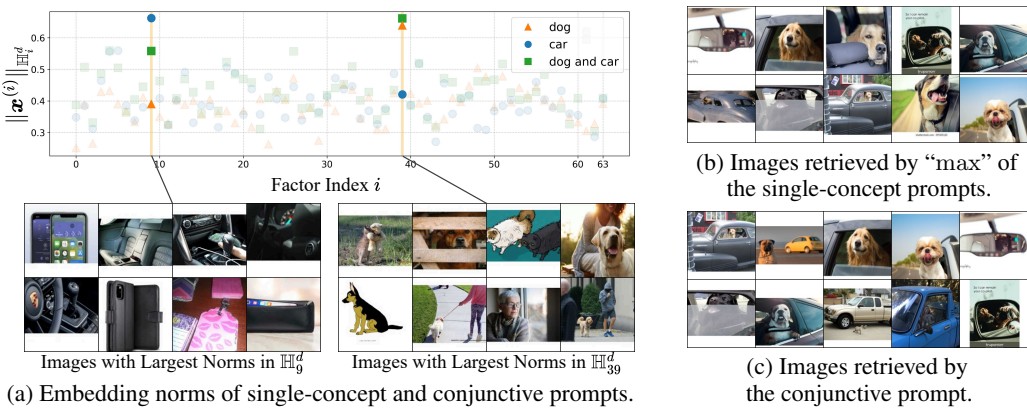

(b) Images retrieved by "max" of the single-concept prompts.

Images with Largest Norms in $\mathbb{H}_9^d$    Images with Largest Norms in $\mathbb{H}_{39}^d$

(a) Embedding norms of single-concept and conjunctive prompts.

(c) Images retrieved by the conjunctive prompt.

Figure 4: **Factor-wise embeddings and retrievals.** (a) Single-concept prompts (e.g., "a dog" or "a car") activate distinct factors (i.e., $i = 39$ or $i = 9$), and their textual composition (e.g., "a dog and a car") activates the corresponding factors simultaneously. (b)–(c) "max" of the single-concept prompts retrieves images similarly to the textual composition. See also Fig. 6 in Appendix D.2.

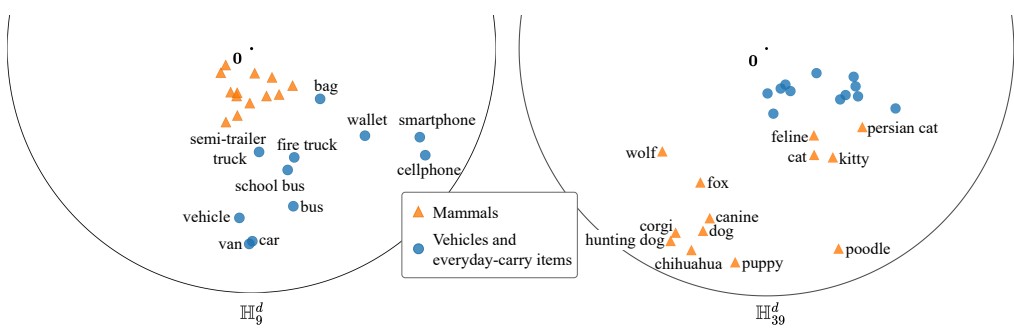

$\mathbb{H}_9^d$        $\mathbb{H}_{39}^d$

Figure 5: **Embeddings projected onto 2D disks by HoroPCA.** A set of relevant concepts (hyponyms of mammals or words related to vehicles and everyday-carry items) forms a hierarchical structure in the corresponding factor ($i = 39$ or $i = 9$), while the same concepts cluster near the origin in another.

origin than "dog." These patterns indicate that factor $i = 39$ encodes the *is-a* taxonomy of mammals (or more specifically, Carnivora). In contrast, in factor $i = 9$, the same embeddings concentrate near the origin, suggesting that PHyCLIP does not use this factor to represent mammals. Embeddings related to vehicles and everyday-carry items exhibit the opposite pattern; they form a hierarchical arrangement in factor $i = 9$ and cluster near the origin in factor $i = 39$.

**Composition via $\ell_1$-Product Metric.** To examine the behavior of the $\ell_1$-product metric, we also obtain the embeddings of the textual composition (e.g., "a photo of a dog and a car"). Figure 4 (a) shows that such a prompt produces large norms in *both* factors $i = 39$ and $i = 9$, meaning that composing concepts simultaneously activates the corresponding factors. Furthermore, we took the factor-wise "max" of the embeddings of two single-concept prompts and found that it retrieves images similarly to the textual composition, as shown in (b)–(c) (see Appendix D.2 for details). These patterns align with the behavior of a Boolean algebra, where multiple concepts are specified by the union of concept subsets (or the element-wise max of binary indicators).

We observe the same pattern for "boy and bicycle" and "sunset and ocean." Consistent with Theorem 2, these observations empirically support that PHyCLIP organizes intra-family taxonomies within individual hyperbolic factors and expresses cross-family compositionality via the simultaneous activation of multiple factors, analogous to a Boolean algebra. We emphasize that, while we provide hierarchy between images, texts, and their cropped versions, we do not provide any explicit supervision for factor assignments or hierarchy between atomic concepts (i.e., words); the specialization of factors and hierarchical structures of atomic concepts emerge automatically through training.

## 5 RELATED WORK

**Vision–Language Models and Representation Learning.**   Vision–language representation learning contributes to retrieval (Mori et al., 1999), semantic segmentation (Barnard et al., 2003), and image generation (Ramesh et al., 2021; Labs, 2025). Early works learned alignments through object-level, word-based classification and detection (Karpathy & Li, 2015; He & Peng, 2017; Engilberge et al., 2018) or through text–image generation (Peng & Qi, 2019; Gu et al., 2018), but they often required complex annotation and network designs (Zhao et al., 2022). A more generic approach maps an entire image or text to a single vector and learns a shared embedding space with a contrastive objective. Representative systems include DeViSE (Frome et al., 2013), VSE++ (Faghri et al., 2018), CLIP (Radford et al., 2021), and ALIGN (Jia et al., 2021). Our model, PHyCLIP, follows this line, while implicitly extracting individual concepts through the geometry of an $\ell_1$-product metric space.

**Hyperbolic Representations in Deep Learning.**   Data often exhibit hierarchical, tree-like structures. Many approaches have attempted to encode such structure (Nguyen et al., 2017; Vulic & Mrksic, 2018), and hyperbolic spaces have become influential due to their empirical performance and theoretical support (Sala et al., 2018; Sonthalia & Gilbert, 2020; van Spengler & Mettes, 2025). As discussed in Section 2, tree metrics admit quasi-isometric embeddings into the two-dimensional hyperbolic plane, which enhances generalization and interpretability (Bridson & Haefliger, 1999; Sarkar, 2011). Hyperbolic embeddings have been applied to words (Nickel & Kiela, 2017; 2018; Tifrea et al., 2019), sentences (Dhingra et al., 2018), graphs (Liu et al., 2019), and images (Khrulkov et al., 2020; Atigh et al., 2022; van Spengler et al., 2023; Qiu et al., 2024). There is also extensive work on building neural networks on hyperbolic spaces (Ganea et al., 2018b; Shimizu et al., 2021; Takeuchi et al., 2022; Peng et al., 2022) and on optimization over Riemannian manifolds (Bonnabel, 2013; Bécigneul & Ganea, 2019). Within vision–language learning, MERU adapts CLIP to hyperbolic geometry (Desai et al., 2023). Our method also leverages hyperbolic geometry and embeds tree-like structures efficiently.

For non-hierarchical data, Euclidean, hyperspherical, or toroidal geometries can be effective (Ebisu & Ichise, 2018), and several studies explore representations in a Riemannian ($\ell_2$) product of such spaces as mixed-curvature representations (Gu et al., 2019; Wang et al., 2024; Gao et al., 2025). PHyCLIP also employs a product space, but all factors are hyperbolic and the product metric is $\ell_1$; we justified both choices theoretically in Section 2.

**Region-based Embeddings for Structured Representations.**   Hierarchical relations can be viewed as a form of inclusion relations. Order embeddings (Vendrov et al., 2016) represent an instance as an upper orthant of Euclidean space, and box embeddings (Vilnis et al., 2018) represent it as an axis-aligned hyperrectangle, where the inverse of set inclusion encodes the hierarchical relation. Euclidean variants include Gaussian embeddings (Vilnis & McCallum, 2015), and hyperbolic variants include disk embeddings (Nickel & Kiela, 2018) and hyperbolic entailment cones (Ganea et al., 2018a). These approaches have also been employed in the vision–language setting (Ren et al., 2016; Desai et al., 2023; Pal et al., 2025). We summarize their theoretical connections in Appendix A.2. These region-based approaches permit composition via intersection of regions, which allows multiple parents and richer semantic composition. However, their compositional expressivity has not yet been fully characterized. We showed in Section 2 that order embeddings and PHyCLIP support compositionality at the level of a Boolean algebra, while a single hyperbolic space may not.

## 6 CONCLUSION

We introduced PHyCLIP, a vision–language model that learns representations using an $\ell_1$-product metric space of hyperbolic factors. We theoretically and empirically demonstrated that it simultaneously captures compositionality across concept families through the $\ell_1$-product metric, as well as *is-a* taxonomies within hyperbolic spaces via hyperbolic embeddings. This design yields state-of-the-art performance across various downstream tasks and provides an interpretable embedding structure. While the main focus is on object composition, it also performs well for attribute binding because it decouples intra-family taxonomy from cross-family composition. By contrast, the relational structure remains unexplored; incorporating its algebraic structure is a promising direction for future work.

## ACKNOWLEDGMENTS

This study was partly supported by JST BOOST (JPMJBY24H0) and CREST (JPMJCR24Q5), and partly achieved through the use of SQUID at D3 Center, Osaka University.

## ETHICS STATEMENT

This study is purely focused on vision–language representation learning, and it is not expected to have any direct negative impact on society or individuals.

## REPRODUCIBILITY STATEMENT

The environment, datasets, methods, evaluation metrics, and other experimental settings are provided in Section 4 and Appendix C. For full reproducibility, the source code is attached as supplementary material.

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

## A  BACKGROUND THEORY

### A.1  QUASI-ISOMETRIC EMBEDDINGS

We adopt standard notions from Bridson & Haefliger (1999). Let $(X, d_X)$ be a metric space. A geodesic segment $[x, y] \subset X$ is an isometric image of an interval whose endpoints are mapped to $x$ and $y$ in $X$. The space $X$ is *geodesic* if every pair of points can be joined by a geodesic segment. For $\delta \geq 0$, a geodesic triangle is $\delta$-*slim* if each side is contained in the $\delta$-neighborhood of the union of the other two sides. A geodesic metric space $X$ is $\delta$-*hyperbolic* (in the sense of Gromov) if every geodesic triangle in $X$ is $\delta$-slim. A metric tree is a geodesic metric space in which any two nodes are joined by a unique geodesic, and every geodesic triangle is a tripod; hence, it is $0$-hyperbolic. Any tree with positive edge lengths, equipped with path length as distance, is a metric tree. Euclidean spaces $\mathbb{R}^n$ are not Gromov-hyperbolic for $n \geq 2$, whereas hyperbolic spaces $\mathbb{H}^n$ are $\delta$-hyperbolic, where $\delta$ depends only on the curvature.

**Definition 2** (Quasi-isometric embedding (Bridson & Haefliger, 1999)). *Let $(X, d_X)$ and $(Y, d_Y)$ be metric spaces. A map $f : (X, d_X) \rightarrow (Y, d_Y)$ is a $(\lambda, c)$-quasi-isometric embedding if it is injective and there exist a distortion $\lambda \geq 1$ and an error $c \geq 0$ such that*

$$\tfrac{1}{\lambda} d_X(\boldsymbol{x}, \boldsymbol{x}') - c \; \leq \; d_Y\big(f(\boldsymbol{x}), f(\boldsymbol{x}')\big) \; \leq \; \lambda \, d_X(\boldsymbol{x}, \boldsymbol{x}') + c \quad \text{for all } \boldsymbol{x}, \boldsymbol{x}' \in X. \tag{7}$$

*If $\lambda = 1$ and $c = 0$, the embedding is* isometric.

### A.2  REPRESENTATIONS OF POSET AND LATTICE

A *poset* $(P, \preceq)$ is a set $P$ with a reflexive, antisymmetric, and transitive relation $\preceq$. Its Hasse diagram places an edge from $x$ to $y$ when $x \prec y$ and there is no $z$ such that $x \prec z \prec y$; hence the existence of an upward path from $x$ to $y$ implies $x \preceq y$ (Ganter & Wille, 1999; Davey & Priestley, 2002). Given $x, y \in P$, a *meet* $x \sqcap y$ (greatest lower bound) and a *join* $x \sqcup y$ (least upper bound) may or may not exist. A *meet-semilattice* (*join-semilattice*) is a poset in which the meet $x \sqcap y$ (the join $x \sqcup y$) exists for all pairs $x, y$, and a *lattice* has both for all pairs.

If a rooted tree is ordered by the ancestor relation with the root $o$ at the bottom (so that $o \preceq x$ for all $x$), then the meet $x \sqcap y$ of a pair $x, y$ always exists, while joins need not exist. Hence, a rooted tree is naturally a meet-semilattice in this orientation. Conversely, an *is-a* taxonomy often uses the *entailment* order $x \preceq y$ interpreted as "$x$ entails $y$" or more roughly "$x$ is more specific than $y$," with the root $o$ at the top. The join $x \sqcup y$ always exists, while meets need not exist; the poset is then a join-semilattice.

Let $\mathcal{C} = \{c_1, \ldots, c_n\}$ be $n$ atomic concepts (e.g., dog, car, tomato,...). A subset $S \subseteq \mathcal{C}$ expresses the conjunction or co-occurrence of the concepts specified in $S$. We define the *Boolean lattice* $(2^{\mathcal{C}}, \subseteq)$ over the power set $2^{\mathcal{C}}$ of $\mathcal{C}$, in which the order relation $\preceq$ is the inclusion relation $\subseteq$. Meet/join are given by intersection/union, respectively. $T \subseteq S$ means that $S$ specifies all concepts in $T$, so $S$ entails $T$. Let $\chi : 2^{\mathcal{C}} \rightarrow \{0, 1\}^n$ be the indicator map with $\chi(S)_i = 1$ iff $c_i \in S$. Then, $T \preceq S$ iff $\chi(T)_i \leq \chi(S)_i$ for all $i$, and meet/join become bit-wise AND/OR, respectively. We summarize the correspondence between different representations in Table 5. In this lattice, each node is defined *intensionally* as a set of concepts.

From the dual perspective, each node can be defined *extensionally* as a set of instances that contain specified concepts, in the context of formal concept analysis (Ganter & Wille, 1999). Let $\mathcal{Z}$ be a universe of instances and let $I \subseteq \mathcal{Z} \times \mathcal{C}$ be an incidence relation (i.e., $z \, I \, c$ means that $z$ has concept $c$). For $S \subseteq \mathcal{C}$, define an operation $S' = \{z \in \mathcal{Z} \mid z \, I \, c \text{ for all } c \in S \subseteq \mathcal{C}\}$, which forms a Galois connection: $S \subseteq T$ implies $T' \subseteq S'$. Also, subsets $S' \subseteq \mathcal{Z}$ form the dual lattice of $(2^{\mathcal{C}}, \subseteq)$, where $S \subseteq T \Leftrightarrow S \preceq T \Rightarrow T' \subseteq S' \Leftrightarrow T' \preceq S'$. If $S = S''$ for any subset $S \subseteq \mathcal{C}$, $S \subseteq T \Leftrightarrow T' \subseteq S'$.

An *is-a* taxonomy is typically realized as a join-subsemilattice of this dual lattice. Order embeddings (Vendrov et al., 2016) can be regarded as an extension of the Boolean lattice, where each bit $\{0, 1\}$ is replaced with a real number $\mathbb{R}$. They declare "$x$ entails $y$" iff $x_i \geq y_i$ for all $i$, similarly to the indicators of a Boolean lattice. Indeed, the ambient poset $(\mathbb{R}^n, \preceq)$ of order embeddings is a lattice with meet/join given by coordinate-wise $\max/\min$, respectively. When regarding an embedding $x$ as an orthant $U(x) \subseteq \mathbb{R}^n$, the entailment is represented as $U(x) \subseteq U(y)$, similarly to the dual lattice. When treating the orthant $U(y)$ as the set of all instances that contain the specified concepts

Table 5: Correspondence of generalization, specialization, and entailment in different representations.

| | Generalization (hypernymy) | Specialization (hyponymy) | Space | Entailment ($\boldsymbol{x}$ or $S$ entails $\boldsymbol{y}$ or $T$) |
|---|---|---|---|---|
| **Tree of *is-a* Relations (*is-a* Taxonomy)** | join $\sqcup$ | (meet $\sqcap$) | $T$ | $\boldsymbol{x} \preceq \boldsymbol{y}$ |
| **Order Embedding (as points)** | min | max | $\mathbb{R}^n$ | $x_i \geq y_i$ for all $i$ |
| **Order Embedding (as orthants)** | | | orthants in $\mathbb{R}^n$ | $U(\boldsymbol{x}) \subseteq U(\boldsymbol{y})$ |
| **Order Embedding (for entailment)** | | | orthants in $\mathbb{R}^n$ | $\boldsymbol{x} \in U(\boldsymbol{y})$ |
| **Hyperbolic Entailment Cone** | (union $\cup$) | intersection $\cap$ | cones in $\mathbb{H}^n$ | $\boldsymbol{x} \in C(\boldsymbol{y})$ |
| **Boolean Lattice (as a power set)** | intersection $\cap$ | union $\cup$ | $2^{\mathcal{C}}$ | $S \supseteq T$ |
| **Boolean Lattice (as a lattice)** | meet $\sqcap$ | join $\sqcup$ | | $S \succeq T$ |
| **Boolean Lattice (with indicator)** | AND | OR | $\{0,1\}^{|\mathcal{C}|}$ | $\chi(S)_i \geq \chi(T)_i$ for all $i$ |
| **Dual Lattice (as a set)** | union $\cup$ | intersection $\cap$ | | $S' \subseteq T'$ |
| **Dual Lattice (as a lattice)** | join $\sqcup$ | meet $\sqcap$ | | $S' \preceq T'$ |
| **Product of Trees** | join $\sqcup$ | (meet $\sqcap$) | $\prod_{i=1}^k T_i$ | $\boldsymbol{x}^{(i)} \preceq \boldsymbol{y}^{(i)}$ for all $i$ |
| **PHyCLIP** | (union $\cup$) | intersection $\cap$ | cones in $(\mathbb{H}_i^d)^k$ | $\boldsymbol{x}^{(i)} \in C_i(\boldsymbol{y}^{(i)})$ for all $i$ |

$\boldsymbol{y}$, the entailment is represented as $\boldsymbol{x} \in U(\boldsymbol{y})$, which aligns with the definition of the dual lattice. Hyperbolic entailment cones (Ganea et al., 2018a) are a hyperbolic extension of the last interpretation of order embeddings, where an orthant $U(\boldsymbol{y})$ is replaced with a geodesic conical region $C(\boldsymbol{y})$.

Also, our proposed PHyCLIP can be regarded as an extension of a Boolean lattice, where each bit $\{0,1\}$ is replaced with a metric tree $T_i$, which is embedded into a hyperbolic factor $\mathbb{H}_i^d$.

# B  PROPOSITIONS, THEOREMS, AND PROOFS

## B.1  PROOF OF PROPOSITION 1

Let $(2^{\mathcal{C}}, \subseteq)$ be a Boolean lattice over all subsets of atomic concepts $\mathcal{C} = \{c_1, \ldots, c_n\}$. The indicator $\chi$ maps subsets $S, T \subseteq \mathcal{C}$ to binary sequences $\chi(S), \chi(T) \in \{0,1\}^n$, where $\chi(S)_i = 1$ if $c_i \in S$ and $\chi(S)_i = 0$ otherwise. Then, $S \subseteq T$ iff $\chi(S)_i \leq \chi(T)_i$ for all $i$. The Hamming distance $d_{\text{Ham}}$ is defined as $d_{\text{Ham}}(\chi(S), \chi(T)) = \sum_{i=1}^n |\chi(S)_i - \chi(T)_i|$. Consider a map $f : \{0,1\}^n \to \mathbb{R}^n, \chi(S) \mapsto \boldsymbol{x} = (x_1, \ldots, x_n) = (1 - \chi(S)_1, \ldots, 1 - \chi(S)_n)$ and the product order $\boldsymbol{x} \preceq \boldsymbol{y}$ iff $x_i \geq y_i$ for all $i$ on $\mathbb{R}^n$. Then, the map $f \circ \chi$ embeds the Boolean lattice $(2^{\mathcal{C}}, \subseteq)$ into the poset $(\mathbb{R}^n, \preceq)$ used by order embeddings while preserving the order relations.

By definition, the Hamming distance is $d_{\text{Ham}}(\chi(S), \chi(T)) = \|\chi(S) - \chi(T)\|_1 = \sum_{i=1}^n |\chi(S)_i - \chi(T)_i|$. Hence, the metric space $(\{0,1\}^n, d_{\text{Ham}})$ is equivalent to an $\ell_1$-product metric space $(\{0,1\}^n, \sum_{i=1}^n |\cdot|)$. Consider a map $f_i$ that maps 0 to the base point of the metric space $X_i$ and 1 to a point with a finite non-zero distance $1/\tau_i > 0$ from the base point. The map $f_i$ is an isometric embedding from $\{0,1\}$ to $X_i$ when scaled by $\tau_i$. The map $f = (f_1, \ldots, f_n)$ is an isometric embedding from $(\{0,1\}^n, d_{\text{Ham}})$ to $(\prod_{i=1}^k \tau_i X_i, \sum_{i=1}^k d_{X_i})$ for any $k \geq n$.

Assume by contradiction that an isometric embedding $f : (\{0,1\}^n, d_{\text{Ham}}) \to (\mathbb{H}^d, d_{\mathbb{H}^d})$ exists for some $n \geq 2$ and $d \geq 2$. Take four points

$$A = (0,0,0,\ldots), \quad B = (1,0,0,\ldots), \quad C = (1,1,0,\ldots), \quad D = (0,1,0,\ldots).$$

in $\{0,1\}^n$. Let $\boldsymbol{a} = f(A), \boldsymbol{b} = f(B), \boldsymbol{d} = f(D)$, and $\boldsymbol{c} = f(C)$. Since $f$ is an isometric embedding,

$$d_{\mathbb{H}^d}(\boldsymbol{a}, \boldsymbol{b}) = d_{\mathbb{H}^d}(\boldsymbol{b}, \boldsymbol{c}) = d_{\mathbb{H}^d}(\boldsymbol{a}, \boldsymbol{d}) = d_{\mathbb{H}^d}(\boldsymbol{d}, \boldsymbol{c}) = 1$$

and

$$d_{\mathbb{H}^d}(\boldsymbol{a}, \boldsymbol{b}) + d_{\mathbb{H}^d}(\boldsymbol{b}, \boldsymbol{c}) = d_{\mathbb{H}^d}(\boldsymbol{a}, \boldsymbol{d}) + d_{\mathbb{H}^d}(\boldsymbol{d}, \boldsymbol{c}) = d_{\mathbb{H}^d}(\boldsymbol{a}, \boldsymbol{c}) = 2.$$

In a hyperbolic space, a geodesic segment is unique, and its midpoint is unique, so both $\boldsymbol{b}$ and $\boldsymbol{d}$ are placed at the midpoint in the geodesic segment $[\boldsymbol{a}, \boldsymbol{c}]$; hence $\boldsymbol{b} = \boldsymbol{d}$. See Proposition I.4 in Bridson & Haefliger (1999). This contradicts the assumption that $f$ is an isometric embedding (which is injective).

## B.2  PROPOSITION 2 AND ITS PROOF

**Proposition 2** ($\ell_1$-product of trees is not hyperbolic). *Let $T_1, T_2$ be infinite metric trees with known bounds for maximum degree and minimum edge length. Their $\ell_1$-product metric space $(T_1 \times T_2, d_1)$*

*is not $\delta$-hyperbolic for any finite $\delta$. Consequently, there is no $(\lambda, c)$-quasi-isometric embedding* $(T_1 \times T_2, d_1) \to \mathbb{H}^n$.

A quasi-geodesic $q$ in $X$ is a $(\lambda, c)$-quasi-isometric embedding $q : I \to X$, where $I$ is an interval in $\mathbb{R}$ or the intersection of $\mathbb{Z}$ with such an interval; see Definition I.8.22 in Bridson & Haefliger (1999). In a $\delta$-hyperbolic space $Y$, the stability of quasi-geodesics asserts that the Hausdorff distance between a geodesic $\gamma$ and a $(\lambda, c)$-quasi-geodesic $q$ with common endpoints is bounded by a constant $D = D(\lambda, c, \delta)$; see Theorem III.1.7 in Bridson & Haefliger (1999).

**Lemma 1** (Stability of geodesic triangles under quasi-isometric embeddings). *Let $X$ be a geodesic metric space and $f : X \to Y$ be a $(\lambda, c)$-quasi-isometric embedding into a $\delta$-hyperbolic space $Y$. Then every geodesic triangle in $X$ is $\tilde{\delta}$-slim for some constant $\tilde{\delta} \leq \lambda(\delta + 2D + c)$, where $D = D(\lambda, c, \delta)$ is the quasi-geodesic stability constant in $Y$.*

*Proof.* Let $\Delta$ be a geodesic triangle in $X$. Each side maps to a $(\lambda, c)$-quasi-geodesic in $Y$. By the stability of quasi-geodesics, each image side is contained in $D$-neighborhood of the corresponding geodesic. Geodesic triangles in $Y$ are $\delta$-slim; hence, each point on one image side is contained in $\delta + 2D$-neighborhood of the union of the other two image sides. Pulling this back via the quasi-isometry inequalities yields the stated bound. $\square$

Let $T_1, T_2$ be infinite trees with bounds for maximum degree and minimum edge length, which admit a geodesic ray of infinite length. For simplicity, we restrict the edge length to be 1, but the following discussion holds for arbitrary non-zero edge lengths, by replacing $\mathbb{N}$ with the ordered set of the geodesic distances from the root to the nodes in the geodesic ray.

Consider $(\mathbb{N}^2, \| \cdot \|_1)$. Let $m$ be an even integer and take three points $A = (0, 0)$, $B = (m, 0)$, $C = (0, m)$. The midpoint $(\frac{m}{2}, \frac{m}{2})$ of a monotone geodesic from $B$ to $C$ is at $\frac{m}{2}$ from $[A, B] \cup [A, C]$, requiring $\delta \geq m/2$. $\delta \to \infty$ as $m \to \infty$. Hence, $(\mathbb{N}^2, \| \cdot \|_1)$ is not $\delta$-hyperbolic for any finite $\delta$.

Choose two geodesic rays $\gamma_i : \mathbb{N} \to T_i$ for $i = 1, 2$. The map $\Phi : \mathbb{N}^2 \to T_1 \times T_2$, $\Phi(m, n) = (\gamma_1(m), \gamma_2(n))$ is an isometric embedding from $(\mathbb{N}^2, \| \cdot \|_1)$ into $(T_1 \times T_2, d_1)$. Given a $\tilde{\delta}$-slim geodesic triangle $\Delta$ in $\mathbb{N}^2$, its image $\Phi(\Delta)$ is also a $\tilde{\delta}$-slim geodesic triangle in $T_1 \times T_2$. Since $(\mathbb{N}^2, \| \cdot \|_1)$ is not $\delta$-hyperbolic, neither is $(T_1 \times T_2, d_1)$.

Assume by contradiction that $f : (T_1 \times T_2, d_1) \to \mathbb{H}^n$ is a $(\lambda, c)$-quasi-isometric embedding, where $\mathbb{H}^n$ is $\delta$-hyperbolic for a finite $\delta$. By Lemma 1, every geodesic triangle in $T_1 \times T_2$ is $\tilde{\delta}$-slim, where $\tilde{\delta} \leq \lambda(\delta + 2D + c)$ and $D = D(\lambda, c, \delta)$ are constants. However, $(T_1 \times T_2, d_1)$ is not $\tilde{\delta}$-hyperbolic for any finite $\tilde{\delta}$, which contradicts the assumption. Therefore, there is no $(\lambda, c)$-quasi-isometric embedding $f : (T_1 \times T_2, d_1) \to \mathbb{H}^n$.

### B.3 PROOF OF THEOREM 2

**Lemma 2** (Product of quasi-isometric embeddings). *If $f_i : (X_i, d_{X_i}) \to (Y_i, d_{Y_i})$ are $(\lambda_i, c_i)$-quasi-isometric embeddings, then*

$$f = \prod_{i=1}^k f_i : \left( \prod_{i=1}^k X_i, \sum_{i=1}^k d_{X_i} \right) \longrightarrow \left( \prod_{i=1}^k Y_i, \sum_{i=1}^k d_{Y_i} \right) \tag{8}$$

*is $(\lambda, c)$-quasi-isometric with $\lambda = \max_i \lambda_i$ and $c = \sum_i c_i$.*

*Proof of Lemma 2.* Sum the index-wise inequalities and bound $\lambda$ by $\max_i \lambda_i$. $\square$

Theorem 2 follows immediately from Theorem 1 and Lemma 2.

## C IMPLEMENTATION DETAILS

### C.1 LORENTZ MODEL OF HYPERBOLIC SPACE

Let $\mathbb{R}^{d,1}$ be the $(d + 1)$-dimensional Minkowski space, equipped with the Minkowski metric $g_{\mathbb{R}^{d,1}} = -\mathrm{d}x_0^2 + \mathrm{d}x_1^2 + \cdots + \mathrm{d}x_d^2$ in coordinates $\hat{\boldsymbol{x}} = (x_0, x_1, \ldots, x_d)$. Intuitively, $x_0$ denotes the time

coordinate, and the others $\boldsymbol{x} = (x_1, \ldots, x_d) \in \mathbb{R}^d$ denote the space coordinates. The inner product in $\mathbb{R}^{d,1}$ is given by

$$\langle \hat{\boldsymbol{x}}, \hat{\boldsymbol{y}} \rangle_{\mathbb{R}^{d,1}} = -x_0 y_0 + \langle \boldsymbol{x}, \boldsymbol{y} \rangle_{\mathbb{R}^d}. \tag{9}$$

For $\alpha > 0$, define the upper sheet of the two-sheeted hyperboloid as $\mathbb{L}_\alpha^d = \{\hat{\boldsymbol{x}} \in \mathbb{R}^{d,1} \mid \langle \hat{\boldsymbol{x}}, \hat{\boldsymbol{x}} \rangle_{\mathbb{R}^{d,1}} = -\alpha^{-1},\ x_0 > 0\}$. Equivalently, every point satisfies $x_0 = \sqrt{\alpha^{-1} + \|\boldsymbol{x}\|_{\mathbb{R}^d}^2}$. The Riemannian metric on $\mathbb{L}_\alpha^d$ is the restriction of the Minkowski metric $g_{\mathbb{R}^{d,1}}$ to $T\mathbb{L}_\alpha^d$; with this metric, the sectional curvature is the constant $-\alpha$ (Cannon et al., 1997; Lee, 2018). The geodesic distance is

$$d_{\mathbb{L}_\alpha^d}(\hat{\boldsymbol{x}}, \hat{\boldsymbol{y}}) = \alpha^{-1/2} \operatorname{arccosh}(-\alpha \langle \hat{\boldsymbol{x}}, \hat{\boldsymbol{y}} \rangle_{\mathbb{R}^{d,1}}) \quad \text{for } \hat{\boldsymbol{x}}, \hat{\boldsymbol{y}} \in \mathbb{L}_\alpha^d. \tag{10}$$

Then, a $d$-dimensional hyperbolic space $\mathbb{H}_\alpha^d$ with a curvature $-\alpha$ is isometrically embedded into $\mathbb{L}_\alpha^d$ by

$$\iota : \mathbb{H}_\alpha^d \to \mathbb{L}_\alpha^d, \boldsymbol{x} \mapsto \hat{\boldsymbol{x}} = (\sqrt{\alpha^{-1} + \|\boldsymbol{x}\|_{\mathbb{R}^d}^2}, \boldsymbol{x}), \tag{11}$$

and we denote $\langle \boldsymbol{x}, \boldsymbol{y} \rangle_{\mathbb{H}_\alpha^d} = \langle \hat{\boldsymbol{x}}, \hat{\boldsymbol{y}} \rangle_{\mathbb{L}_\alpha^d}$ and $d_{\mathbb{H}_\alpha^d}(\boldsymbol{x}, \boldsymbol{y}) = d_{\mathbb{L}_\alpha^d}(\hat{\boldsymbol{x}}, \hat{\boldsymbol{y}})$ in the main body.

When feature extractors (such as encoders) operate in the Euclidean space $\mathbb{R}^d$, their output cannot be treated directly as an embedding $\boldsymbol{x}$ in a hyperbolic space due to the mismatch in geometry. Instead, the output $\boldsymbol{v} = (v_1, \ldots, v_d)$ is treated as a tangent vector in the tangent space $T_{\hat{\boldsymbol{o}}}\mathbb{L}_\alpha^d \simeq \mathbb{R}^d$ at the base point $\hat{\boldsymbol{o}} = (\alpha^{-1/2}, 0, \ldots, 0)$ of $\mathbb{L}_\alpha^d$ and mapped to a point in $\mathbb{L}_\alpha^d$ via the exponential map

$$\exp_{\hat{\boldsymbol{o}}}^\alpha : T_{\hat{\boldsymbol{o}}}\mathbb{L}_\alpha^d \to \mathbb{L}_\alpha^d, \boldsymbol{v} \mapsto \hat{\boldsymbol{x}} = \exp_{\hat{\boldsymbol{o}}}^\alpha(\boldsymbol{v}) = \cosh(\sqrt{\alpha}\|\boldsymbol{v}\|_{\mathbb{R}^d})\hat{\boldsymbol{o}} + \frac{\sinh(\sqrt{\alpha}\|\boldsymbol{v}\|_{\mathbb{R}^d})}{\sqrt{\alpha}\|\boldsymbol{v}\|_{\mathbb{R}^d}}\boldsymbol{v}. \tag{12}$$

### C.2 Hyperbolic Entailment Cones in the Lorentz Model

Hyperbolic entailment cones capture the hierarchical relationships (Ganea et al., 2018a). For every point $\boldsymbol{y}$ in each hyperbolic factor $\mathbb{H}^d$, we define a geodesic conical region $C(\boldsymbol{y})$, where all points $\boldsymbol{x} \in C(\boldsymbol{y})$ are considered more specific than $\boldsymbol{y}$ (i.e., $\boldsymbol{x} \preceq \boldsymbol{y}$). The size of this conical region is determined by its half-aperture $\omega(\boldsymbol{y})$, which is inversely proportional to the norm:

$$\omega(\boldsymbol{y}) = \sin^{-1}\left(\min\left\{1, \frac{2K}{\sqrt{\alpha}\|\boldsymbol{y}\|_{\mathbb{R}^d}}\right\}\right), \tag{13}$$

where $K$ is set to 0.1. Then, $\boldsymbol{x} \in C(\boldsymbol{y})$ iff $\phi(\boldsymbol{x}, \boldsymbol{y}) < \omega(\boldsymbol{y})$ for the exterior angle

$$\phi(\boldsymbol{x}, \boldsymbol{y}) = \cos^{-1}\left(\frac{x_0 + y_0 \alpha \langle \boldsymbol{x}, \boldsymbol{y} \rangle_{\mathbb{H}_\alpha^d}}{\|\boldsymbol{y}\|_{\mathbb{R}^d}\sqrt{(\alpha\langle \boldsymbol{x}, \boldsymbol{y} \rangle_{\mathbb{H}_\alpha^d})^2 - 1}}\right) \tag{14}$$

### C.3 Model Architecture and Hyperparameters

We introduce the details of our implementation and hyperparameters, which follow Desai et al. (2023); Pal et al. (2025) unless specified otherwise.

As an image encoder, we employ the Vision Transformer (Dosovitskiy et al., 2021; Chen et al., 2021; Touvron et al., 2021) with a patch size of 16. Each image is randomly resized by a scale from 0.5 to 1.0 and randomly cropped to $224 \times 224$ pixels, resulting in 196 tokens, concatenated with 2-D sine–cosine position embeddings. We employ the text encoder used by the original CLIP (Radford et al., 2021), which consists of a 12-layer Transformer architecture (Vaswani et al., 2017) with embeddings of 512 dimensions.

The outputs of image and text encoders are scaled by learnable scalars $c_{\text{img}}$ and $c_{\text{txt}}$, respectively, before being mapped by the exponential map. These scalars are initialized to $c_{\text{img}} = c_{\text{txt}} = 1/\sqrt{512}$. The negative curvature $\alpha_i$ for factor $i$ is initialized at 1.0 and clamped in $[0.1, 10.0]$. For the contrastive loss $\mathcal{L}_{\text{cont}}$ in Eq. (3), the temperature $\tau$ is initialized to 0.07 and clipped at a minimum value of 0.01. For the entailment loss $\mathcal{L}_{\text{ent}}$ in Eq. (5), the hyperparameter $\eta$ is set to $\eta = 0.7$ for inter-modality entailments ($I \preceq T$ and $I^{\text{box}} \preceq T^{\text{box}}$) and $\eta = 1.2$ for intra-modality entailments ($T \preceq I^{\text{box}}$ and $T \preceq T^{\text{box}}$). These scalars are learned on a logarithmic scale.

The hyperparameter $\gamma$ for the overall loss in Eq. (1) is set to $\gamma = 0.2$. We trained each model on 4 A100 GPUs for 500,000 iterations with a batch size of 768. For the large Vision Transformer, we used 8 A100 GPUs. We used the AdamW optimizer (Loshchilov & Hutter, 2019) with hyperparameters $\beta_1 = 0.9, \beta_2 = 0.98$. We applied weight decay of 0.2 to model parameters but not to scalar parameters. We used a cosine learning rate scheduler (Loshchilov & Hutter, 2017) with a maximum learning rate of $5 \times 10^{-4}$ and a warm-up of 4,000 steps.

We found that the learned curvatures differ across factors and sometimes reach the upper bound 10.0 or the lower bound 0.1, but we did not observe a consistent pattern across factors.

## C.4 BENCHMARKS

**Zero-shot Image Classification.** We follow the protocol in Desai et al. (2023). Each class is accompanied by a set of short text templates, such as "a photo of a {class name}". The prediction is made by selecting the class whose text templates are closest on average to the image in the embedding space. We summarize the datasets below.

- **ImageNet** (Russakovsky et al., 2015): A large-scale dataset of diverse, everyday object categories.

- **Food-101** (Bossard et al., 2014): A fine-grained dataset of 101 different types of food dishes.

- **CIFAR-10** (Krizhevsky & Hinton, 2009): A dataset of low-resolution natural images across 10 general object classes.

- **CIFAR-100** (Krizhevsky & Hinton, 2009): Similar to CIFAR-10, but with 100 fine-grained object classes.

- **CUB-2011** (Wah et al., 2011): A fine-grained dataset for the identification of 200 bird species.

- **SUN397** (Xiao et al., 2010): A large-scale scene recognition dataset with 397 scene categories.

- **Stanford Cars** (Krause et al., 2013): A fine-grained dataset of cars, annotated with make, model, and year.

- **FGVC-Aircraft** (Maji et al., 2013): A fine-grained dataset for aircraft model recognition.

- **DTD** (Cimpoi et al., 2014): The Describable Textures Dataset for texture recognition.

- **Oxford-IIIT Pets** (Parkhi et al., 2012): A fine-grained dataset of 37 different pet breeds.

- **Caltech-101** (Li et al., 2004): One of the classic object recognition datasets with 101 categories.

- **Flowers-102** (Nilsback & Zisserman, 2008): A fine-grained dataset for the classification of 102 flower categories.

- **STL-10** (Coates et al., 2011): An image recognition dataset inspired by CIFAR-10, but with higher resolution.

- **EuroSAT** (Helber et al., 2019): A dataset of Sentinel-2 satellite images for land use and land cover classification.

- **RESISC45** (Cheng et al., 2017): A benchmark for Remote Sensing Image Scene Classification (RESISC).

- **Country211** (Radford et al., 2021): A dataset for predicting the country of origin from photographs.

**Zero-shot Image and Text Retrieval.** In text-to-image retrieval, given a text query, the model retrieves the nearest images in the embedding space, and vice versa in image-to-text retrieval. Please refer to the detailed protocol in Desai et al. (2023). We summarize the datasets used as follows.

- **COCO** (Lin et al., 2014): A large-scale dataset of complex everyday scenes with rich annotations.

- **Flickr30K** (Young et al., 2014; Karpathy & Li, 2015): A dataset of images from the Flickr website, each paired with five descriptive captions.

Table 6: Results with different model sizes.

| | | w/ boxes | Hierarchical Classification | | | | | VL-CheckList–Object | | | | | |
| --- | --- | --- | --- | --- | --- | --- | --- | --- | --- | --- | --- | --- | --- |
| | | | WordNet | | | | | Location | | | Size | | |
| | | | TIE($\downarrow$) | LCA($\downarrow$) | $J(\uparrow)$ | $P_H(\uparrow)$ | $R_H(\uparrow)$ | Center | Mid | Margin | Large | Medium | Small |
| **ViT S/16** | CLIP | | 4.126 | 2.445 | 0.7530 | 0.8289 | 0.8314 | 61.83 | 62.23 | 60.33 | 63.30 | 61.23 | 59.07 |
| | MERU | | 4.189 | 2.438 | 0.7486 | 0.8278 | 0.8265 | 60.90 | 59.20 | 58.00 | 62.20 | 60.00 | 58.70 |
| | HyCoCLIP | ✓ | **3.669** | **2.231** | **0.7818** | **0.8514** | **0.8507** | 69.67 | 68.17 | 67.67 | 72.33 | 66.33 | 68.10 |
| | **PHyCLIP** | ✓ | 3.715 | 2.241 | 0.7778 | 0.8492 | 0.8476 | **73.33** | **71.37** | **72.00** | **75.27** | **68.23** | **70.77** |
| **ViT B/16** | CLIP | | 3.750 | 2.276 | 0.7774 | 0.8471 | 0.8483 | 61.90 | 60.30 | 60.40 | 63.87 | 60.17 | 58.23 |
| | CLIP | ✓ | 3.736 | 2.279 | 0.7784 | 0.8473 | 0.8501 | 61.93 | 59.33 | 60.83 | 63.70 | 60.80 | 58.07 |
| | MERU | | 3.815 | 2.294 | 0.7733 | 0.8454 | 0.8450 | 61.27 | 59.03 | 58.97 | 63.97 | 57.70 | 56.07 |
| | MERU | ✓ | 3.802 | 2.289 | 0.7740 | 0.8457 | 0.8455 | 61.03 | 58.47 | 58.73 | 62.63 | 58.70 | 56.47 |
| | HyCoCLIP | ✓ | 3.319 | 2.092 | 0.8043 | 0.8676 | 0.8661 | 70.43 | 69.50 | 67.80 | 72.57 | 66.13 | 67.20 |
| | **PHyCLIP** | ✓ | **3.294** | **2.083** | **0.8059** | **0.8684** | **0.8672** | **71.20** | **70.30** | **70.37** | **73.73** | **68.10** | **67.83** |
| **ViT L/16** | CLIP | | 3.480 | 2.173 | 0.7956 | 0.8595 | 0.8613 | 63.33 | 60.63 | 59.87 | 64.57 | 59.50 | 58.03 |
| | MERU | | 3.571 | 2.191 | 0.7891 | 0.8565 | 0.8551 | 62.10 | 58.00 | 57.23 | 63.93 | 57.83 | 54.57 |
| | HyCoCLIP | ✓ | 3.113 | 2.012 | 0.8175 | 0.8769 | 0.8753 | 73.03 | 69.83 | 70.23 | 73.77 | 67.53 | 68.43 |
| | **PHyCLIP** | ✓ | **3.038** | **1.991** | **0.8226** | **0.8797** | **0.8793** | **73.60** | **70.90** | 70.07 | 73.73 | **68.07** | **69.27** |

Among methods with the same backbone, the best and second-best performances are emphasized by bold fonts and underlines, respectively.

**Hierarchical Classification.** This task was introduced in Russakovsky et al. (2015), and we used the implementation in Pal et al. (2025). The class labels are enriched by WordNet (Miller, 1995), and the embeddings of class labels are obtained in the same way as the zero-shot image classification task. Errors between predicted and true classes are measured using the WordNet graph with unit-length edges. Tree Induced Error (TIE) is the distance between the nodes corresponding to predicted and true classes. Lowest Common Ancestor (LCA) error is the maximum of the distances from predicted and true classes to their LCA. Jaccard similarity $J$, hierarchical precision $P_H$, and hierarchical recall $R_H$ are similarities between the sets of ancestors of predicted and true classes. Intuitively, hierarchical precision $P_H$ quantifies correctness under over-generalization: it takes value 1 if the predicted label is the ground truth or one of its ancestors in the taxonomy. Conversely, hierarchical recall $R_H$ quantifies correctness under over-specialization: it takes value 1 if the predicted label is the ground truth or one of its descendants.

**Compositional Understanding.** Samples in typical multi-modal datasets are diverse enough that there are few near-duplicate image–text pairs; consequently, models insensitive to detailed semantics can still perform well on retrieval tasks. To assess whether a model truly understands the compositionality of words in a caption, hard negative captions are generated, which are almost correct but differ in a small, targeted way to evaluate whether models can select the true caption.

In VL-CheckList–Object, nouns in the caption are replaced. Because the difficulty varies with the replaced object's location (center/mid/margin) and size (small/medium/large) in image, results are reported separately for each subset.

In SugarCrepe, three operations (replace, swap, and add) are applied to objects, attributes, and relations. *Replace-Obj* is similar to VL-CheckList–Object. *Swap* exchanges roles or pairings. In *Swap-Obj*, the model must correctly resolve agent–action combinations. *Add* introduces nouns or adjectives that were absent from the original caption.

# D  ADDITIONAL RESULTS AND VISUALIZATIONS

## D.1  ADDITIONAL EXPERIMENTAL RESULTS

We summarized the results of single runs with the small and large Vision Transformers as the image encoder (Dosovitskiy et al., 2021; Chen et al., 2021; Touvron et al., 2021) in Table 6. As the model size increases, the overall performance improves in most cases. Nevertheless, PHyCLIP remains the best or at least competitive across all evaluation metrics for hierarchy and compositionality.

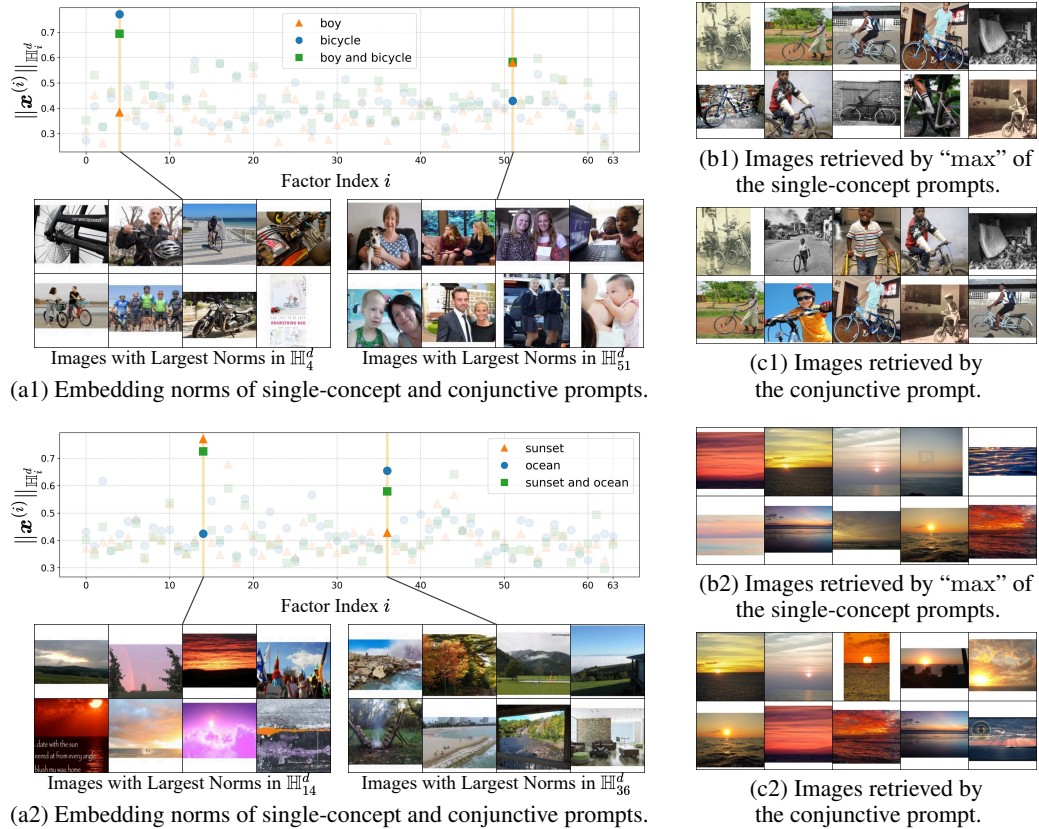

(b1) Images retrieved by "$\max$" of the single-concept prompts.

(c1) Images retrieved by the conjunctive prompt.

(b2) Images retrieved by "$\max$" of the single-concept prompts.

(c2) Images retrieved by the conjunctive prompt.

Images with Largest Norms in $\mathbb{H}_4^d$

Images with Largest Norms in $\mathbb{H}_{51}^d$

(a1) Embedding norms of single-concept and conjunctive prompts.

Images with Largest Norms in $\mathbb{H}_{14}^d$

Images with Largest Norms in $\mathbb{H}_{36}^d$

(a2) Embedding norms of single-concept and conjunctive prompts.

Figure 6: **Factor-wise embeddings and retrievals.** (a1)(a2) Single-concept prompts activate distinct factors, and their textual composition activates the corresponding factors simultaneously. (b1)–(c1), (b2)–(c2) "$\max$" of the single-concept prompts retrieves images similarly to the textual compositions. See also Fig. 4

## D.2 Additional Visualizations

In this section, we provide additional visualizations that complement Section 4.3. We embed each word using the single-concept prompt, "a photo of a {word}" and the conjunctive prompt, which is the textual composition of two words, "a photo of a {word 1} and a {word 2}."

Figure 6 visualizes factor-wise embedding norms of single-concept and conjunctive prompts. In (a1), the "boy" embedding activates factor $i = 51$, which is also activated by various human images, indicating that this factor captures humans; the "bicycle" embedding activates factor $i = 4$, associated with bicycles and wheels. The conjunctive prompt "boy and bicycle" activates both factors $i = 4$ and $i = 51$. In (a2), the "sunset" embedding activates factor $i = 14$, which captures a family of skies, whereas the "ocean" embedding activates factor $i = 36$, which captures a family of natural landscapes. The conjunctive prompt "sunset and ocean" activates both factors $i = 14$ and $i = 36$.

The contrastive loss (InfoNCE loss) uses the distance summed over all factors, so it allows embeddings to gather at the origin in a factor if they are sufficiently distant from each other in another factor. As shown in Eq. (13), the aperture of the hyperbolic entailment cone increases as the distance from the origin decreases, up to a maximum of 180 degrees. Consequently, near the origin, a child instance can be embedded anywhere within half of the space without incurring entailment loss. These mechanisms allow instances to effectively "turn off" some hyperbolic factors. We also considered a modification in which the cone aperture could reach 360 degrees to completely eliminate the penalty, but since the original setting already behaved well in our experiments, we decided not to modify this component.

Figure 4 (b)–(c), Figure 6 (b1)–(c1), and (b2)–(c2) show top-10 GRIT images retrieved using the factor-wise "$\max$" of single-concept prompts and conjunctive prompts. Specifically, we embed two

single-concept prompts (e.g., "a photo of a dog" and "a photo of a car") as $\boldsymbol{X}_a = (\boldsymbol{x}_a^{(1)}, \ldots, \boldsymbol{x}_a^{(k)})$ and $\boldsymbol{X}_b = (\boldsymbol{x}_b^{(1)}, \ldots, \boldsymbol{x}_b^{(k)})$, and then we construct a new embedding $\boldsymbol{X}_{\max\{a,b\}}$ by selecting, for each factor, the factor-wise embedding with the larger norm between two single-concept prompts, i.e., we take

$$\boldsymbol{X}_{\max\{a,b\}} = (\boldsymbol{x}_{\max\{a,b\}}^{(1)}, \ldots, \boldsymbol{x}_{\max\{a,b\}}^{(k)}) \text{ with } \boldsymbol{x}_{\max\{a,b\}}^{(i)} = \underset{\boldsymbol{x} \in \{\boldsymbol{x}_a^{(i)}, \boldsymbol{x}_b^{(i)}\}}{\arg\max} \ \|\boldsymbol{x}\|_{\mathbb{H}_i^d} \text{ for } i = 1, \ldots, k.$$

Then, the factor-wise norms satisfy $\|\boldsymbol{x}_{\max\{a,b\}}^{(i)}\|_{\mathbb{H}_i^d} = \max\{\|\boldsymbol{x}_a^{(i)}\|_{\mathbb{H}_i^d}, \|\boldsymbol{x}_b^{(i)}\|_{\mathbb{H}_i^d}\}$. If each factor were a bit $\{0, 1\}$, this operation would reduce to the union operation or the logical OR for a Boolean algebra. If each factor were a real number $\mathbb{R}$, it coincides with an element-wise max, examined in order embeddings (Vendrov et al., 2016). The retrieval results by both methods are appropriate in most cases and often overlap. Concepts specified in the prompt are embedded with large norms in factors that capture their corresponding concept families, whereas unspecified concepts are represented with small norms. Consequently, by retaining only the high-norm factors, we can compose concepts without corrupting the semantics of the original prompts. These results suggest that PHyCLIP expresses cross-family composition in a manner analogous to Boolean algebra and order embeddings.

## THE USE OF LARGE LANGUAGE MODELS.

We used ChatGPT and GitHub Copilot as assistance tools for polishing the manuscript and implementing the experimental code. We did not use large language models for research ideation or for proofs.

