# OpenReview forum: "PHyCLIP: $\ell_1$-Product of Hyperbolic Factors Unifies Hierarchy and Compositionality in Vision-Language Representation Learning"
_ICLR.cc/2026/Conference — ICLR 2026 Poster_

### Official Review · Reviewer_QmqT · 2025-10-20

**Soundness:** 3
**Presentation:** 3
**Contribution:** 4
**Rating:** 8
**Confidence:** 4

**Summary:**

This paper proposes a new model named PHyCLIP, which extends the HyCoCLIP method to a Cartesian product of hyperbolic spaces to allow for an understanding of both concept hierarchies and concept compositionality. First, the authors provide a theoretical motivation, explaining how this Cartesian product of hyperbolic spaces is capable of embedding these two types of information. This background is followed by an exposition of their PHyCLIP model. With a comprehensive suite of experiments the authors then show that their proposed model leads to significant improvements with respect to baselines such as CLIP, MERU and HyCoCLIP. Moreover, in an interesting analysis they show that the individual hyperbolic factors indeed appear to learn different concept hierarchies.

**Strengths:**

- The paper proposes a solution to an important problem of hierarchical vision-language models such as MERU and HyCoCLIP, which struggle to model both compositionality and hierarchical understanding. The proposed use of Cartesian products of hyperbolic spaces seems elegant and effective, providing an important step in the right direction.
- The theoretical background is clearly presented, well-motivated and forms a nice basis for the rest of the paper.
- The method and its presentation are simple, yet effective.
- The results in the extensive suite of experiments are quite convincing in my opinion.
- To me, the analysis in Subsection 4.3 felt particularly compelling.

**Weaknesses:**

It feels to me that there is a slight disconnect between the theory and the implementation, due to a few minor points:
1. The background (Theorem 2) assumes the existence of some product of trees that is to be embedded into a product of hyperbolic spaces. Here, if my understanding is correct, each tree is considered the replacement of some bit that would normally represent the presence of some concept. However, it is not completely clear from the text how this tree represents the presence or absence of a concept within the composition. Is the distance of the node to the root of the tree some form of the "degree of presence" of this concept? Would absence then be indicated by picking the root node? If so, that might be useful to discuss more explicitly in lines 174-181.
2. After discussing the embedding of this product of trees, the method section dives straight into a presentation of the method. However, it seems to me that we do not actually have any known hierarchies and that we assume the model to learn these implicitly (as discussed shortly at the end of section 4). A short discussion on this point somewhere in or directly after lines 188-195 might be beneficial for the structure of Section 3 as I was not quite sure about the connection of Theorem 2 to the actual proposed method upon reading this section.
3. The proposed loss seems to assign the same type of loss to each hyperbolic factor. However, it seems to me that this would sometimes hamper the learning of hierarchies in specific factors. For example, given an image of a dog next to a car, how should the image box containing just the car be represented in the hyperbolic factor that contains the mammal hierarchy? I would assume that this should be put close to the origin of this factor, but the loss would punish the model for such a representation. This seems like a difficult problem to address through any modifications of the loss and the analysis in 4.3 already shows that the model acts as expected despite this potential problem, but I think a discussion on this limitation should still be included somewhere.

I do want to add that the theory, method and experiments individually are all convincing and that, overall, I consider these weaknesses minor compared to the strengths of the paper.

Lastly, a small remark: in lines 88-89 the authors mention (Sarkar, 2011; Nickel & Kiela, 2017; Ganea et al., 2018a), but given the requirement for low distortion, I think a more appropriate list would be (Sarkar, 2011; Sala et al., 2018; [1]), as the other two were not designed for low distortion and generally do not achieve it.

[1] Max van Spengler, Pascal Mettes. Low-distortion and GPU-compatible Tree Embeddings in Hyperbolic Space. In _International Conference on Machine Learning (ICML)_, 2025.

**Questions:**

Aside from the questions posed in the weaknesses, I have some more small questions:

1. In lines 188-195 it is mentioned that "we use standard hyperbolic embeddings (Nickel & Kiela, 2017) together with hyperbolic entailment cones (Ganea et al., 2018a). It is not completely clear to me what is meant here. Would this be the proposed approach for embedding actual trees or does this reference to the PHyCLIP method's loss presented after?
2. Theorems 1 and 2 both use $\mathbb{H^2}$, but I think this should be $\mathbb{H}^d$, right?

---

> ### Author Response · Authors · 2025-11-19
>
> We sincerely thank the reviewer for devoting their valuable time to carefully reading our paper and for providing many constructive suggestions. We deeply appreciate that you found our design elegant and effective, viewed it as an important step in the right direction, considered the theoretical background to be clearly presented and well motivated, and regarded the experimental results as convincing. We have carefully considered all of your suggestions and will incorporate the corresponding clarifications into the manuscript.
>
> > 1. […] Is the distance of the node to the root of the tree some form of the "degree of presence" of this concept? Would absence then be indicated by picking the root node? […]
>
> Thank you for your important question. In our design, the distance of the node to the root represents ”degree of specialization,” rather than  "degree of presence." The root node states “the existence of some object, while not specifying what it is,” rather than the absence.  A node may state “the existence of a dog, while not specifying what breed it is” and a further node may state “ the existence of a chihuahua”. This design is consistent with the dual to the Boolean lattice in Appendix A.2, where each node is a set of instances that share specified attributes (e.g., dog), and these instances may or may not have attributes that are additionally specified in child nodes (e.g., chihuahua).
>
> We will incorporate this explanation into Section 3.
>
> >  2. […] it seems to me that we do not actually have any known hierarchies and that
> we assume the model to learn these implicitly […]
>
> As you point out, during training, the model is not given explicit information about the hierarchy between concepts, but learns such hierarchy implicitly. The emergence of hierarchical structure between concepts (typically, words) without explicit supervision has already been observed by HyCoCLIP in the context of vision-language models and by Chen et al. [1] in the context of large language models. Motivated by these observations, we focused on designing an embedding space in which hierarchical structures can be represented more naturally.
>
> [1] Chen et al., Probing BERT in Hyperbolic Spaces, ICLR, 2021.
>
> To this end, at the beginning of Section 3, we model the structure of sentences and images as a product of trees, and then show in Theorem 2 that the proposed space is well suited to embedding this structure. Theorem 2 explicitly connects the structural assumptions about sentences and images with our design of geometry. We agree that this motivation should be stated more clearly, and we plan to add this explanation to Section 3.
>
> By contrast, the hierarchy between text and image, as well as the hierarchy between the whole and the part, is provided explicitly by the GRIT dataset and is therefore learned in a supervised manner. We briefly mention this difference in types of hierarchy in Fig. 1, but we will clarify this aspect more explicitly in the revised version.
>
> > 3. […] how should the image box containing just the car be represented in the hyperbolic factor that contains the mammal hierarchy? I would assume that this should be put close to the origin of this factor, but the loss would punish the model for such a representation. […]
>
> Thank you for this sharp observation. As you point out, in a hyperbolic factor that specializes to concepts (e.g., mammals) irrelevant to a given instance (e.g., car), that instance is put close to the origin.
>
> The contrastive loss (InfoNCE loss) uses the distance summed over all factors, so it allows embeddings to gather at the origin in a factor if they are distant from each other in another factor. As shown in Eq. (13) in Appendix, the aperture of the hyperbolic entailment cone increases as the distance from the origin decreases, up to a maximum of 180 degrees. Consequently, near the origin, a child instance can be embedded anywhere within half of the space without incurring entailment loss. These mechanisms allow instances to effectively "turn off" some hyperbolic factors.
>
> We also considered a modification in which the cone aperture could reach 360 degrees to completely eliminate the penalty, but since the original setting already behaved well in our experiments, we decided not to modify this component.
>
> > I do want to add that the theory, method and experiments individually
>  are all convincing and that, overall, I consider these weaknesses minor
>  compared to the strengths of the paper.
>
> Your comments have given us a valuable opportunity to improve the manuscript. We sincerely appreciate it.
>
> > Lastly, […] given the requirement for low distortion, I think a more appropriate list would be
>  (Sarkar, 2011; Sala et al., 2018; [1]) […]
>
> Thank you for bringing this important reference to our attention. We agree that this work should be cited. In the final version, we plan to cite it preferentially in all discussions related to distortion.

---

> > ### Author Response · Authors · 2025-11-19
> >
> > >1. In lines 188-195 it is mentioned that "we use standard hyperbolic embeddings (Nickel & Kiela, 2017) together with hyperbolic entailment cones (Ganea et al., 2018a). It is not completely clear to me what is meant here.  […]
> >
> > By this sentence, we intended to provide the reader with some intuition before the formal definitions in the next paragraph. However, we now realize that introducing the loss function in the middle of a discussion on geometry is indeed abrupt. We believe that simply removing the sentence you pointed out will improve the readability, and we plan to do so. Thank you for this helpful suggestion.
> >
> > > 2. Theorems 1 and 2 both use $\mathbb{H}^2$, but I think this should be $\mathbb{H}^d$, right?
> >
> > We intended to emphasize that these Theorems hold even with $d=2$. However, as you point out, the most accurate statement is for $\mathbb{H}^d$ with $d \ge 2$, and we plan to revise the text accordingly.

---

> > ### Comment · Reviewer_QmqT · 2025-11-21
> >
> > Thank you for your detailed responses! Based on these I have two small follow-up questions:
> >
> > 1. Regarding my first question you mention that the root node, which I assume means the origin, indicates the existence of an object. Do you mean the existence of an object belonging to the hierarchy corresponding to this particular factor or simply an object in general? In case of the prior, can we easily tell from the individual hyperbolic factors whether some hierarchy is completely absent from a scene?
> > 2. The mentioned modification to the aperture where it can reach 360 degrees at the origin seems interesting (although I accept that it is beyond the scope of this paper). Purely out of curiosity, is that something that you have experimented with or something you intend to try in future work?

---

> > > ### Author Response · Authors · 2025-11-23
> > >
> > > > 1. Regarding my first question you mention that the root node [...]
> > >
> > > Thank you for this careful clarification. Our previous explanation was somewhat ambiguous. In our assumption, the root node of a tree (that is, the origin of a hyperbolic factor) indicates simply the existence of an object in general. This allows us to represent the possibility of "the absence of objects belonging to the hierarchy corresponding to this particular factor". Thus, our assumption corresponds to the latter interpretation you described, and under this assumption, we can potentially "tell from the individual hyperbolic factors whether some hierarchy is completely absent from a scene". If we had assumed the prior interpretation, that would have been difficult. Of course, this remains an assumption, and the actual embeddings are slightly displaced from the origin. Nonetheless, the results in Figure 5 indicate that the empirical behavior is broadly consistent with this assumption.
> > >
> > > > 2. The mentioned modification to the aperture [...]
> > >
> > > Sorry for the ambiguity. We have not experimented with this modification. The original implementation employs arcsin, which does not support half-apertures larger than 90 degrees (that is, apertures larger than 180 degrees). Using the aperture of 360 degrees at the origin requires non-trivial changes and tuning, as well as an ablation study. We would very much like to try this in future work.
> > >
> > > We sincerely appreciate your thoughtful comments, which helped us to polish the manuscript.

---

> > > > ### Comment · Reviewer_QmqT · 2025-11-25
> > > >
> > > > Thank you for your responses. I am satisfied with the answers and will retain my rating. Good luck with the rest of the discussion phase.

---

> > > > > ### Author Response · Authors · 2025-11-27
> > > > >
> > > > > Thank you once again for your thoughtful feedback.

---

### Official Review · Reviewer_iHKL · 2025-11-01

**Soundness:** 3
**Presentation:** 3
**Contribution:** 4
**Rating:** 8
**Confidence:** 3

**Summary:**

This paper proposes PHyCLIP  to tackle a central limitation of contrastive vision–language models: representing both hierarchical “is-a” structure within a concept family and cross-family compositionality in a single embedding space. The proposed model PHyCLIP, which embeds images and texts into a Cartesian product of hyperbolic factors and endows that product with an ℓ₁-product metric. The design strives for a clean division of labor: intra-family hierarchies emerge inside individual hyperbolic factors, while cross-family composition is captured by the ℓ₁ aggregation across factors, in analogy with Boolean conjunction. Empirically, PHyCLIP is trained (from scratch) on the GRIT corpus with region-level supervision and evaluated on zero-shot classification, image–text retrieval, hierarchical classification, and compositional understanding; the model is reported to outperform single-space Euclidean or single-space hyperbolic baselines and to yield interpretable factor activations aligned with concept families.

**Strengths:**

To the best of my knowledge, the idea of representing vision–language concepts in a Cartesian product of hyperbolic factors with an ℓ₁-product metric is a clean and original way to reconcile hierarchical “is-a” structure (within factors) with cross-family compositionality (across factors) in the context of vision-language models. I think the theory–architecture fit of this paper is very strong: hyperbolic factors give you room for tree-like taxonomies, while ℓ₁ aggregation behaves like a sparse, Boolean-style conjunction that naturally penalizes missing parts in a composition.

I also like that the learning objective stays simple—contrastive InfoNCE on the product metric (This is consistent with training recipes of common contrastive-based vision language models such as the baselines) plus a hyperbolic entailment-cone regularizers, which, in my view, makes comparisons fair.

Empirically, I also find the evaluation breadth convincing: zero-shot classification and retrieval improve over Euclidean and single-space hyperbolic baselines, hierarchical metrics on ImageNet+WordNet move in the right direction, and compositional checks (e.g., VL-Checklist, SugarCrepe) benefit in the settings the method is designed for.

Overall I think the contribution is solid and this work can benefit the overall community

**Weaknesses:**

I see the main gaps as scope and analysis, not methodology.

First, the paper focuses on object/attribute composition; explicit relational composition (multi-object relations and their algebra) remains largely out of scope, yet those cases are where many Vision–Language systems struggle. It would be interesting to see if there are further experiments on thoes flavors of compositionality

Second, the text-retrieval side is “competitive” rather than consistently superior; If possible a targeted error analysis to separate caption ambiguity from potential factor misalignment would be very benefitial to the ML community.

Third, the method’s behavior as the number of factors k grows deserves a bit more operational guidance: the results suggest a sweet spot, but it is not entirely clear how to pick k for new domains with different taxonomic granularity.

Finally, because training relies on large-scale auto-annotated data, I would appreciate a brief characterization of how annotation noise (e.g., imperfect region phrases) interacts with the entailment-cone loss and with factor specialization—more as insight than as a requirement.

**Questions:**

On the data side I am curious about if the authors observe any systematic sensitivity to noisy region phrases or ambiguous attributes, and could factor-wise uncertainty or gating mitigate this?

---

> ### Author Response · Authors · 2025-11-19
>
> Thank you for devoting your valuable time to reading our paper and for proving the thoughtful and constructive comments. We deeply appreciate that you found the idea clean and original, comparisons fair, and evaluations convincing. We have carefully considered all comments and answer as follows.
>
> >First, the paper focuses on object/attribute composition; explicit relational composition (multi-object relations and their algebra) remains largely out of scope […]
>
> We sincerely acknowledge that the relational composition remains out of scope despite its importance. This limitation is also reflected in Table 3 (SugarCrepe), where PHyCLIP underperforms CLIP on Replace-Rel (replacement of the relation between objects) and Swap-Obj (swapping objects). This fact also suggests that a hybrid between PHyCLIP and the original CLIP may enjoy the strengths of both. We will incorporate this potential future work in the final version. However, we consider that this drawback is sufficiently compensated by its effectiveness on the target multi-object compositionality, as evidenced by 5-9% performance improvements on VL-CheckList-Object.
>
> >Second, […] separate caption ambiguity from potential factor misalignment
>
> We apologize for the insufficient explanation in the original manuscript. VL-CheckList-Object and SugarCrepe are benchmarks based on text retrieval, and their results directly explain why our PHyCLIP is competitive on text retrieval for COCO and Flickr. PHyCLIP is strong for additions and replacements of objects and adjectives, whereas it is weaker for replacements of relations. We believe that the performance variation for COCO and Flickr is due to which types of hard negatives, and in what proportion, are included in each benchmark.
>
> >Third, [..] how to pick k for new domains with different taxonomic granularity.
>
> GRIT is a large and diverse dataset, so the best configuration (k=64, d=8) appears reasonably generic. Importantly, even with fewer factors (e.g., k=8,d=64), PHyCLIP still outperforms the baseline, HyCoCLIP. Thus, the overall improvements are robust to the choice of k, and careful tuning is not essential for PHyCLIP to provide benefits.
>
> >Finally, […] how annotation noise (e.g., imperfect region phrases) interacts
> >Questions: [….] any systematic sensitivity to noisy region phrases or ambiguous attributes
>
> This is a very interesting perspective. We inspected instances in GRIT dataset and did find some instances that appear to be mislabeled, but not enough to reveal a clear pattern. If there were a statistical bias in the mislabels, it could indeed affect the results, but we believe that the larger scale of GRIT, compared to COCO or Flickr, is still an advantage.

---

### Official Review · Reviewer_LzC9 · 2025-11-01

**Soundness:** 2
**Presentation:** 3
**Contribution:** 2
**Rating:** 6
**Confidence:** 3

**Summary:**

The paper proposes PHyCLIP, which employs an ℓ1-product metric on a Cartesian product of hyperbolic factors to jointly capture hierarchical structures within concept families and compositionality across them. While the theoretical motivation is interesting and experiments are comprehensive, critical design choices lack sufficient justification and the improvements are marginal.

**Strengths:**

1. Clear design hypothesis. The paper explicitly assigns intra-family hierarchy to hyperbolic factors and inter-family composition to ℓ1 aggregation. The overview (Fig. 2, p.2) and factor-wise norm plots/activations (Figs. 4–6) align qualitatively with this hypothesis.
2. The method is tested on classification, retrieval, hierarchical metrics, and composition benchmarks. PHyCLIP is generally competitive and often best, with notable gains on VL-CheckList–Object and small but consistent improvements on several other metrics (Tables 1–3).
3. Interpretability hints. Factor-wise visualizations (pp.8, 21–22) show families (e.g., mammals vs. vehicles) specializing to different factors and conjunctions activating multiple factors—matching the intended Boolean-like behavior.
4. Broad evaluation. The method is tested on classification, retrieval, hierarchical metrics, and composition benchmarks. PHyCLIP is generally competitive and often best, with notable gains on VL-CheckList–Object and small but consistent improvements on several other metrics (Tables 1–3).

**Weaknesses:**

1. Unconvincing ℓ1 justification
- Prop. 1 shows Boolean lattices embed in ℓ1, but the model uses continuous hyperbolic factors, not discrete bits
- Missing: Why not weighted combinations or learned aggregation functions?
2. No statistical validation
- Single runs without confidence intervals
- Improvements often 1-2%, likely within noise
3.  Implementation opacity
- Curvature initialization/learning dynamics unexplained
- "ℓ2-product metric" undefined mathematically
4. Marginal gains don't justify complexity
- Average improvement ~1% over HyCoCLIP
- 64× more distance computations
- 64× more curvature parameters
- Memory/compute overhead unreported

**Questions:**

1. Do ℓ1’s gains persist under a full ℓp sweep with ≥3 seeds?
2. How do factorized Euclidean/order/box models with ℓ1 compare at equal total dimension?
3. Is the ℓ2 baseline a true Riemannian product distance implementation? Please provide the formula/code.
4. What is the quality-vs-cost curve as k increases (1, 8, 64, 128)?

---

> ### Author Response · Authors · 2025-11-19
>
> Thank you for your valuable time and for providing the thoughtful and constructive suggestions. We deeply appreciate that you recognized the clear design hypothesis, interpretability, notable gains on VL-CheckList-Object, and the broad evaluation. Below, we address each of your points in detail.
>
> >1. Unconvincing ℓ1 justification
>
> >Prop. 1 shows Boolean lattices embed in ℓ1, but the model uses continuous hyperbolic factors, not discrete bits
>
> Thank you for an important question. By an embedding, we mean a mapping from one space into another richer space (for example, higher-dimensional or equipped with additional operations), while preserving the original structure. In our setting, each discrete bit is embedded into a larger set, namely a continuous hyperbolic factor. Through this embedding, everything that can be represented by a discrete bit (namely, 0 or 1) can still be represented in a hyperbolic factor (as the origin and a point at distance 1 from the origin), and, in addition, new characteristics (in particular, hierarchical structure) become representable.
>
> In addition, among ℓp-product metrics, ℓ1-product metric is the only metric that coincides with the Hamming distance for a Boolean lattice. In this sense, ℓ1-product of hyperbolic factors can be justified as a generalization of a Boolean lattice.
>
> >Why not weighted combinations or learned aggregation functions?
>
> Thank you for this insightful comment. In the current implementation, weighted combinations are already implicitly realized. When the negative curvature $\alpha$ is multiplied by $1/c^2$, the Riemannian metric is scaled by $c^2$, and the distance between any two points becomes $c$ times larger. Hence, learning the negative curvature of each hyperbolic factor effectively corresponds to learning the combination weight.
>
> However, since the distance of each instance's embedding from the origin can be adjusted independently for each instance, a small curvature does not necessarily mean that the corresponding factor is emphasized. If all embedding points lie near the origin, that factor is effectively down-weighted.
>
> > 2. No statistical validation
>
> Following your valuable suggestion, we have started multiple runs in order to report confidence intervals. However, due to limited computational resources, we may not be able to complete all runs and include the results within the rebuttal period. Prior related work, such as HyCoCLIP, reports single-run results, and we trained all models with the same random seed, so we believe that our current reporting is consistent with common practice.
>
> > 3. Implementation opacity
>
> > Curvature initialization/learning dynamics unexplained
>
> Sorry for the lack of clarity. At line 968 in Appendix, we state that the negative curvature is initialized to 1. The learned curvatures differ across factors and sometimes reach the upper bound 10.0, but we did not observe a consistent pattern across factors.
>
> >"ℓ2-product metric" undefined mathematically
>
> Thank you for pointing out. The ℓp-product metric is defined as $d_{\ell_p}( (x^{(1)},\dots,x^{(k)}), (y^{(1)},\dots,y^{(k)}) )= (\sum_{i=1}^k |d_i(x^{(i)}, y^{(i)})|^p)^{1/p}$.
>
> We will replace Definition 1 with this to clarify the definitions of both ℓ1- and ℓ2-product metrics.
>
> > 4.  Marginal gains don't justify complexity
>
> >complexity
>
> >64× more distance computations
>
> >64× more curvature parameters
>
> >Memory/compute overhead unreported
>
> The proposed PHyCLIP requires $k$ evaluations of $\cosh$ and $\sinh$ for the exponential maps and $k$ evaluations of $\operatorname{arccosh}$ for factor-wise distances. These operations are fully parallelized across factors and are needed only once per embedding or distance. In practice, the wall‑clock time is dominated by the Vision Transformer and text encoder, and the additional cost from these operations is negligible at the scale of our backbones.
>
> PHyCLIP introduces only k-1=63 additional parameters to learn curvatures, which is negligible compared to the 86M parameters for the base Vision Transformer.

---

### Official Review · Reviewer_Rpq7 · 2025-11-03

**Soundness:** 2
**Presentation:** 3
**Contribution:** 2
**Rating:** 4
**Confidence:** 3

**Summary:**

Hyperbolic embeddings capture hierarchy and partial orders via inclusion. These approaches have been used to train vision-language models such as MERU and HyCoCLIP, which capture concept hierarchies. On the other hand, these hyperbolic embeddings lack a canonical composition operation. To solve this problem, the paper introduces PHyCLIP, which uses an $l_{1}$ product metric to unify hierarchy and composition of concepts. Results on zero-shot classification, image-text retrieval, and compositional benchmarks show that PHyCLIP outperforms methods using hyperbolic representations.

**Strengths:**

The paper is well written. The related work section is comprehensive and up to date.

The method is simple to implement. The work builds on HyCoCLIP and shows that using an $l_{1}$ product to unify the hierarchy and composition of concepts improves results across all benchmark tasks.

**Weaknesses:**

**Missing Baseline.**
The paper does not compare PHyCLIP to CLIP fine-tuned on the GRIT dataset. Furthermore, fine-tuning CLIP could, in fact, improve performance on the compositionality benchmark tasks. In SugarCrepe [a], Table 5 shows much higher performance for fine-tuned CLIP. This is a strong reason to include fine-tuned CLIP as a baseline for fair comparison.

**Compositionality.**
Table 3 shows improvements for PHyCLIP, but not by a significant margin over CLIP (approximately 1.4 points on average), and only marginal improvements over CLIP on SugarCrepe. This is not a strong result for a method that aims to improve compositionality in vision-language models.

**Sensitivity to k.**
In Table 4, changing the number of factors (k) can significantly reduce performance. However, there isn’t a clear guideline for choosing this hyperparameter.


**References.**

[a] SugarCrepe: Fixing Hackable Benchmarks for Vision-Language Compositionality. NeurIPS 23.

**Questions:**

See weaknesses.

---

> ### Author Response · Authors · 2025-11-19
>
> Thank you for devoting your valuable time to reading our paper and for providing the thoughtful comments. We sincerely appreciate your recognition of the comprehensive related work section and the improved performance of the proposed method. We are grateful for your constructive feedback and address your concerns as follows.
>
> > **Missing Baseline.** The paper does not compare PHyCLIP to CLIP fine-tuned on the GRIT dataset.[…]
>
> Thank you for an interesting suggestion. Although CLIP fine-tuned on GRIT may improve absolute performance, this direction is orthogonal to our primary focus on embedding geometry. Our goal is to assess how the geometry of the embedding space affects the performances of CLIP variants. To isolate this factor, we trained all models (CLIP, MERU, HyCoCLIP, and PHyCLIP) from scratch on the same GRIT dataset, where the only difference is the geometry of the embedding space. This design yields a controlled and fair comparison for our research question.
>
> >**Compositionality.** […] only marginal improvements over CLIP on SugarCrepe. […]
>
> The form of compositionality we target is multi-object co-occurrence (for example, whether dog and car appear together in an instance). This is exactly what VL-CheckList-Object evaluates, and on this benchmark, PHyCLIP yields substantial improvements (+5-9%). Even though the total improvements on SugarCrepe are indeed marginal, the pattern that PHyCLIP is strong on the targeted form of compositionality (e.g., Replace-Obj, Add-Obj) and weaker on another form of compositionality (e.g., replacements of relations between objects, evaluated by Replace-Rel, Swap-Obj) is consistent with the intended role of our geometric design. The latter form of compositionality is also important and a promising direction for future research.
>
> >**Sensitivity to k.** […]
>
> Thank you for an important question. GRIT is a large and diverse dataset, so the best configuration (k=64, d=8) appears reasonably generic. Importantly, even with fewer factors (e.g., k=8, d=64), PHyCLIP still outperforms the baseline, HyCoCLIP. Thus, the overall improvements are robust to the choice of k, and careful tuning is not essential for PHyCLIP to provide benefits.

---

### Author Response · Authors · 2025-12-04

Dear Reviewers,

We would like to sincerely thank all reviewers for carefully reading our paper and giving us positive and insightful feedback. We are grateful that you recognized the importance of our problem setting, and that you found our theoretical background and empirical results promising.

Your comments and questions have provided us with a valuable opportunity to refine the explanations and eliminate ambiguities, making the final version easier to understand.

Once again, we sincerely appreciate the time and effort you devoted to reviewing our work.

Sincerely,

Authors

---

### Meta-Review · Area_Chair_HQEC · 2026-01-03

**Summary:**

The decision for this paper was exceptionally difficult and represents a classic borderline case. On one hand, two reviewers (scores of 8) strongly advocated for the work, praising its "theory-architecture fit" and its ability to offer more interpretable embedding structures. On the other hand, a more skeptical camp (scores of 4 and 6) pointed to significant empirical fragile points.

Ultimately, the recommendation for acceptance is driven by the conceptual originality and elegance of the proposed PHyCLIP framework. By endowing a Cartesian product of hyperbolic factors with an $\ell_1$-product metric, the authors provide a novel approach that addresses an important challenge in vision-language models: modeling intra-family hierarchies and cross-family compositionality.

Two reviewers (scores of 8) strongly advocated for the work, praising its "theory-architecture fit" and its ability to offer more interpretable embedding structures. While a more skeptical camp (scores of 4 and 6) pointed to insufficient empirical evidence—specifically the marginal magnitude of gains and the lack of statistical significance testing—the meta-reviewer concludes that the architectural novelty and the potential for a new geometric paradigm in representation learning outweigh the current empirical limitations. The work serves as a valuable "proof of concept" for unifying Boolean-like logic with hyperbolic taxonomies.

**Reviewer Concerns:**

Concerns Addressed by the Rebuttal

* Theoretical intuition (`QmqT`): The authors successfully explained how the root node/origin represents "existence" and distance represents "specialization."

* Complexity argument (`LzC9`): The authors correctly pointed out that the additional parameter count (63 curvature scalars) is negligible compared to the total model size.

Outstanding Concerns

* Missing baseline comparisons (`Rpq7`): While the authors argued that training from scratch on GRIT is fair, the lack of a comparison to the well-established, standard CLIP fine-tuned on this specific data leaves a gap in understanding whether the geometric shift is truly a significant step forward.

* Lack of statistical significance (`LzC9`): Reviewer `LzC9` specifically requested multiple runs and confidence intervals. The authors admitted they could not complete these within the rebuttal period, which is understandable given the limited time. However, given that the reported improvements are often as low as 1% over the baseline, it is impossible to determine if the results are truly the result of the proposed geometry or simple random seed variance.

* Failure on relational compositionality (`Rpq7`, `iHKL`): The model specifically underperforms or matches standard CLIP on SugarCrepe for relational tasks (e.g., swapping objects). This suggests that the $\ell_1$ "Boolean" logic work for a very narrow subset of compositionality (presence/absence of objects), calling into question the broader claims made in the paper.

**Reviewer Scores:**

In light of the outstanding concerns listed above, if the discussion phase had allowed for a deeper dive into the empirical data, this meta-reviewer believes that the scores would likely have shifted as follows:

* Reviewer `Rpq7` (Initial: 4 $\rightarrow$ Estimated: 4): The reviewer’s primary concern was the marginal gains (1.4 points on SugarCrepe) and the lack of a well-tuned baseline. Since the authors' response provided no new baseline results for a fine-tuned CLIP on GRIT and did not provide a definitive guideline for the hyperparameter $k$, the reviewer’s skepticism would have remained firmly at a 4.

* Reviewer `LzC9` (Initial: 6 $\rightarrow$ Estimated: 5): This reviewer was the most vocal regarding statistical rigor. The authors' explicit admission that they could not provide multiple runs or confidence intervals within the rebuttal period is a critical failure for a paper claiming a 1% improvement, making the results indistinguishable from noise.

* Reviewer `iHKL` (Initial: 8 $\rightarrow$ Estimated: 7): Initially a champion of the work. While they would likely acknowledge the empirical noise and relational failure, their appreciation for the "theory-architecture fit" would likely keep them in the acceptance range.

* Reviewer `QmqT` (Initial: 8 $\rightarrow$ Estimated: 8): This reviewer explicitly stated post-rebuttal: "I am satisfied with the answers and will retain my rating." Their focus remained on the elegant resolution of theoretical ambiguities.

---

### Decision · Program_Chairs · 2026-01-26

Accept (Poster)